# Modelling CO₂ and N₂O emissions from soils in silvopastoral systems of the West-African Sahelian band

Yélognissè Agbohessou[1,2,3], Claire Delon[4], Manuela Grippa[5], Eric Mougin[5], Daouda Ngom[1], Espoir Koudjo Gaglo[1,3], Ousmane Ndiaye[2,6], Paulo Salgado[7,8], Olivier Roupsard[3,9,10]

[1]Université Cheikh Anta Diop, Département de Biologie Végétale, Dakar, Senegal
[2]Institut Sénégalais de Recherches Agricoles, Dakar, Senegal
[3]LMI IESOL, Centre IRD-ISRA de Bel Air, Dakar, Senegal
[4]Laboratoire d'Aérologie, Université de Toulouse, CNRS, IRD, UPS, Toulouse, France
[5]Géosciences Environnement Toulouse, Université de Toulouse, CNES, CNRS, IRD, UPS, Toulouse, France
[6]Centre de Recherches Zootechniques de Dahra, Institut Sénégalais de Recherches Agricoles, Dahra, Senegal
[7]CIRAD, UMR SELMET, F-34090 Montpellier, France
[8]UMR SELMET, University of Montpellier, CIRAD, INRAE, Institut Agro, F-34090 Montpellier, France
[9]CIRAD, UMR Eco&Sols, Dakar, Senegal
[10]Eco&Sols, Univ Montpellier, CIRAD, INRAE, Institut Agro, IRD, Montpellier, France

*Correspondence to*: Yélognissè Agbohessou, (yelognissefredi.agbohessou@ucad.edu.sn)

**Abstract.** Silvopastoral systems (SPSs) have been shown to improve ecosystem resilience and provide sustainable land management solutions in the Sahel. However, accurately estimating the contribution of Sahelian ecosystems to the overall greenhouse gas (GHG) balance is a challenge, in particular the magnitude of carbon dioxide ($CO_2$) and nitrous oxide ($N_2O$) emissions from soils. In this work, we spatialized and applied the process-based model STEP-GENDEC-$N_2O$ to investigate the magnitude, spatial, and temporal patterns of herbaceous mass, as well as $CO_2$ and $N_2O$ emissions from soil (not net emissions) in Sahelian SPSs. Our results show that over the last decade (2012-2022), there was a heterogeneous spatial distribution of herbaceous mass production, as well as of soil $CO_2$ and $N_2O$ emissions in Sahelian SPSs. Spatial variations in soil $CO_2$ emissions are primarily controlled by soil carbon content, temperature, herbaceous mass, and animal load, while soil nitrogen content, soil water content, and animal load are the main factors driving the spatial variations in $N_2O$ emissions from soil. The estimated $CO_2$ and $N_2O$ emissions from soil in Sahelian SPSs over the 2012-2022 period were equal to $58.79 \pm 4.83$ Tg $CO_2$-C yr$^{-1}$ (1 Tg = $10^{12}$ g) and $21.59 \pm 3.91$ Gg $N_2O$-N yr$^{-1}$ (1 Gg = $10^9$ g), respectively. These values are generally lower than estimates reported in the literature for tropical areas and croplands. Furthermore, our simulations indicated a significant annual rising trend of soil $CO_2$ and $N_2O$ emissions between 2012-2020 as herbaceous mass increases, making more C and N available for nitrification, denitrification and decomposition processes. By mapping soil $CO_2$ and $N_2O$ emissions, we provide crucial insights into the localization of emission hotspots in Sahelian SPSs, thereby offering valuable information that can be used to devise and implement effective strategies aimed at fostering carbon sequestration in the Sahel.

Keywords: Sahelian Silvopastoral system, soil carbon dioxide emission, soil nitrous oxide emission, process-based model, greenhouse gas emission

# 1 Introduction

Carbon dioxide ($CO_2$) and nitrous oxide ($N_2O$) are two important greenhouse gases (GHG) that contribute significantly (>90%) to anthropogenic climate warming (Hansen et al., 2000). With 298 times the warming potential of $CO_2$ over 100 years (Myhre et al., 2013), $N_2O$ is also a stratospheric ozone-depleting substance (Ravishankara et al., 2009). The atmospheric concentrations of $CO_2$ and $N_2O$ have experienced significant increases since the late 1700s (Bloch-Johnson et al., 2021; Prinn et al., 2018). This surge is primarily attributed to emissions originating from terrestrial soils (Butterbach-Bahl et al., 2013;

Chevallier et al., 2015; Tian et al., 2020) during the period from 1700 to 1980 (Kammen and Marino, 1993). However, post-1990, the major contributors to greenhouse gas emissions on a global scale shifted to the energy systems and industrial sectors (Parmesan et al., 2022). $CO_2$ emissions from soil are due to organic matter decomposition (Robertson and Paul, 2000), while $N_2O$ is produced in soils through nitrification (*i.e.,* oxidation of ammonium to nitrate) and denitrification (*i.e.,* reduction of nitrate to molecular N) (Davidson and Verchot, 2000). These processes are regulated by a range of environmental factors

(Aulakh et al., 1991; Bajracharya et al., 2000; Reth et al., 2005), making it difficult to up-scale soil $CO_2$ and $N_2O$ emissions from local sites to the regional and global scale.

Nevertheless, in the last decade, several works provided estimates of $CO_2$ and $N_2O$ emissions from terrestrial soils at the large scale (Dangal et al., 2020; Leahy, 2004; Tian et al., 2020, 2019, 2018, 2016, 2015). However, regions such as Africa, especially West-African Sahelian livestock production systems have not received much attention. Our knowledge of the magnitude, and

the spatio-temporal distribution of soil $CO_2$ and $N_2O$ emissions in these systems is limited and subject to large uncertainties (Assouma et al., 2017). This is mainly due to a lack of experimental and modelling studies focused on the region.

Silvopastoral systems (SPSs) are one of the most common livestock production systems in the West-African Sahel (Le Houerou, 1987; Herrero et al., 2013b, a; Turner et al., 2014). They are composed of a mix of trees and herbaceous cover, grazed by livestock. As an attractive nature-based climate solution, SPSs offer long-term climate benefits thanks to the presence

of trees that have the potential to sequester carbon and offset GHG emissions (Agbohessou et al., 2023; Torres et al., 2017). On the other hand, it has been reported that the livestock component of SPSs has an impact on the nitrogen (N) and carbon (C) cycles and therefore on GHG emissions (Butterbach-Bahl et al., 2020). Indeed, livestock affects substrate availability in soil through N input from their excreta, then impacting $CO_2$ and $N_2O$ emissions (Butterbach-Bahl et al., 2020; Dangal et al., 2020). It has been also reported that direct agricultural $N_2O$ emissions from Africa mainly arise from livestock manure deposited in

pastures and rangelands (Xu et al., 2019). Livestock movements result in heterogeneous spatial and temporal distributions of excreta, which increases spatial heterogeneity in soil properties and available nutrients which promote microbiological processes driving soil $CO_2$ and $N_2O$ emissions (Assouma et al., 2017; Smith et al., 2003). Actually, rangeland soils, combined with livestock productions, were reported to be responsible for a large share of GHG emissions (Assouma et al., 2017; Soussana et al., 2010; Valentini et al., 2014). The importance of rangelands in the global $CO_2$ and $N_2O$ cycles and their potentialities to

increase atmospheric $CO_2$ and $N_2O$ levels, have been highlighted in a number of studies (Chang et al., 2015; Dangal et al.,

2020; Leahy, 2004). Accordingly, to better understand the magnitude of GHG emissions in these systems and develop effective and spatially targeted climate solutions it is important to identify $CO_2$ and $N_2O$ emission hotspots and accurately estimate emissions from Sahelian SPSs.

The different bottom-up approaches used to estimate large-scale soil $CO_2$ and $N_2O$ emissions include the use of "emission factors" (EFs) as proposed by the Intergovernmental Panel on Climate Change (IPCC) (Hergoualc'h et al., 2019; IPCC, 2006), statistical extrapolation of field measurements, and process-based models (Bigaignon et al., 2020; Delon et al., 2019; Li et al., 2000; Parton et al., 2001). Besides, the top-down approaches integrate atmospheric measurements and atmospheric inversion models (Saikawa et al., 2014). Each method has its uncertainties and limitations, resulting in significant divergences in results across studies (Tian et al., 2019), especially in underrepresented regions like West-Africa (Tian et al., 2020). The IPCC defined $N_2O$ emission as 1% of the applied N in the Tier 1 level (IPCC, 2006). This assumption of constant EF can neither depict spatial variations in $N_2O$ emissions nor reflect the impacts of changing environments over time (Davidson and Kanter, 2014). Statistical extrapolation can also fail to depict the spatial heterogeneity in emissions, especially when the spatial variability in the parameters exceeds the conditions prevailing during the calibration step (Tian et al., 2019). On the other hand, the process-based model simulation approach has the advantage of describing the overall C and N cycle within the terrestrial systems and can integrate various driving factors controlling soil $CO_2$ and $N_2O$ production and emissions (Tian et al., 2019). This approach involves the use of extensive data, such as meteorological, soil and ecosystem management data. However, estimating the model parameters can be challenging as there is a scarcity of experimental studies that encompass comprehensive details on local and regional pedoclimatic conditions and agricultural practices in West-Africa. Additionally, reliable and accurate large spatial scale input datasets for the models are often lacking, not only in under-represented areas but also in well-documented regions like Europe (Ballabio et al., 2016).

In this study, we selected the STEP-GENDEC-$N_2O$ process-based model (Agbohessou et al., 2023), which couples water budget, herbaceous aboveground and belowground vegetation growth and decay, herbaceous and tree foliage litterfall (Jarlan et al., 2005; Mougin et al., 1995; Tracol et al., 2006), soil biogeochemistry and gaseous emissions (Bigaignon et al., 2020; Delon et al., 2019; Moorhead and Reynolds, 1991) to investigate the spatial and temporal patterns of herbaceous vegetation mass, $CO_2$ and $N_2O$ emissions from soil, and estimate their annual budget in the Sahelian SPSs. The STEP-GENDEC-$N_2O$ model was specifically designed for Sahelian semi-arid ecosystems and has been validated locally for soil $CO_2$ and $N_2O$ emissions in several sites representative of the Sahelian SPSs (Agbohessou et al., 2023; Bigaignon et al., 2020; Delon et al., 2019, 2015). In this study, this model was upscaled and used at the regional scale, *i.e.* at the west Sahelian region scale.

The specific objectives of our study are to: (1) investigate the spatio-temporal patterns of herbaceous vegetation mass, $CO_2$ and $N_2O$ emissions from soils in the Sahelian SPSs over the last decade (2012-2022); (2) identify the environmental factors responsible for the changes in the spatial patterns of soil $CO_2$ and $N_2O$ emissions; and (3) estimate the soil $CO_2$ and $N_2O$ budget of the Sahelian SPSs during the 2012-2022 period.

## 2 Materials and methods

### 2.1 Characteristics of the study area

The Sahel region is a semi-arid strip stretching across the African continent from Senegal to the red sea (Le Houérou, 1989). The region is characterized by high temperatures, low soil fertility and a long dry season alternating with a short rainy season, with precipitation occurring mostly between June and September, making it challenging to grow crops. As a result, a large portion of the region is used for pastoral activities, which serve as the primary means of subsistence (Touré et al., 2012). The focus of this study is put on Sahelian SPSs of West-Africa from longitude 18°W to 20°E and latitude 13°N to 18°N (Fig. 1

and A1), which cover approximately 40% ($\approx$ 892,353 km$^2$) of the Sahelian band. The dynamics of rainfall in the Sahel are strongly linked to the dynamics of the West African monsoon (Biasutti, 2019). The Sahel experienced a dry period from the late 1960s to the mid-90ies, marked by years of extreme droughts such as in 1973-1974 and 1984-1985. Several studies have reported a recovery period (Galle et al., 2018; Nicholson, 2017) for the Sahel since 1984, which is defined by an increasing trend in total seasonal rainfall (Biasutti, 2019; Dai et al., 2004). However, rainy season characteristics have changed; rainfall

is more intense and intermittent (especially in areas with the lowest rainfall) and wetting is concentrated in the late rainy season (Biasutti, 2019; Chagnaud et al., 2022).

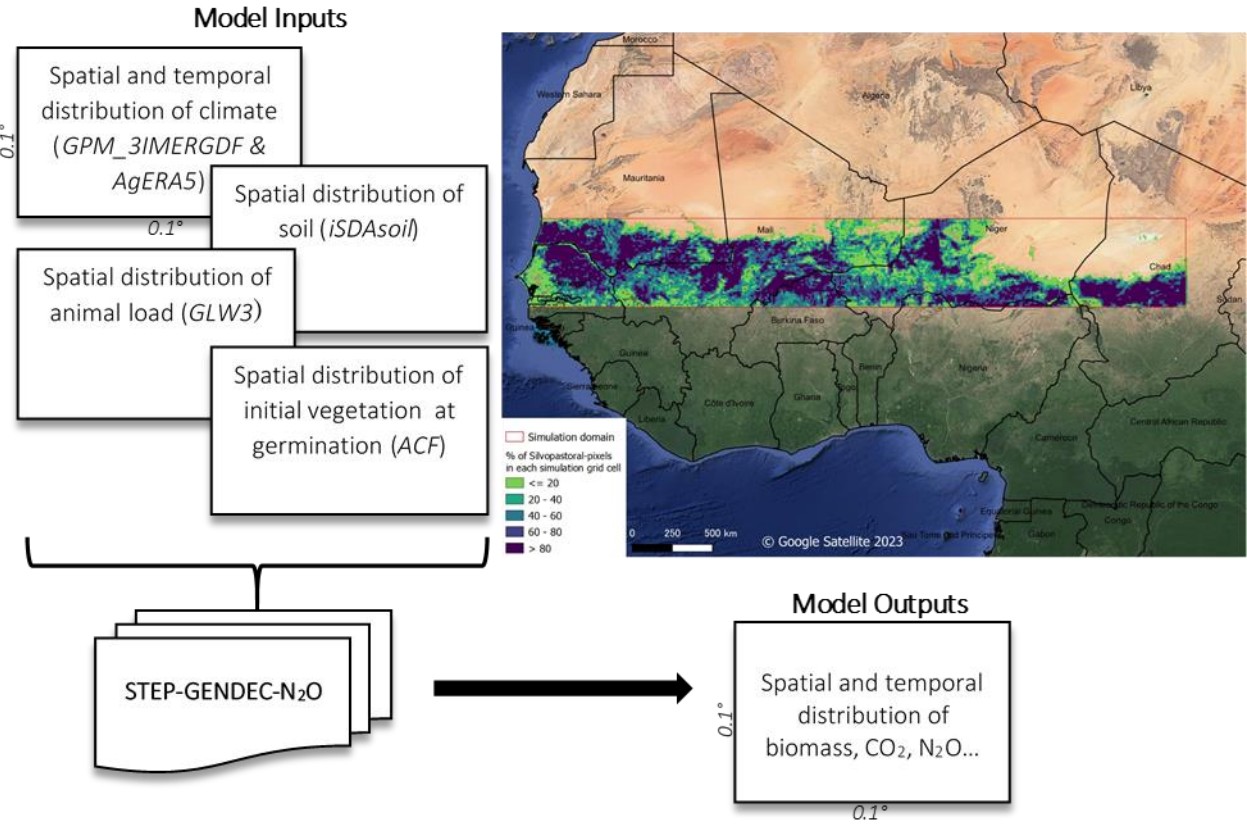

**Figure 1: Illustration of the up-scaling approach used: model inputs and outputs and the simulation domain (Sahelian SPSs) are shown on the map. Silvopastoral areas were filtered from cultivated areas in the simulation area.**

## 2.2 Model used from 1D processes to 2D up scaling: STEP-GENDEC-N₂O

### 2.2.1 Model description

STEP-GENDEC-N$_2$O is a process-based model developed for the Sahelian herbaceous savanna, coupling water budget, aboveground and belowground herbaceous vegetation growth and decay, litter fall (Mougin et al., 1995), soil biogeochemistry (Moorhead and Reynolds, 1991), and soil gaseous emissions (Agbohessou et al., 2023; Bigaignon et al., 2020; Delon et al., 2019). The model simulates the main processes describing the water, C, and N cycling between the atmosphere, vegetation, and soil at daily time steps and finally CO$_2$ and N$_2$O emissions. STEP-GENDEC-N$_2$O is forced daily by rain, global radiation, air temperature, wind speed, and relative air humidity. The model has been applied and evaluated to estimate herbaceous vegetation mass in Senegal, Mali (Mougin et al., 1995; Tracol et al., 2006), and Niger (Hiernaux et al., 2009), CO$_2$, NO, and N$_2$O emissions in Mali (Delon et al., 2015) and Senegal (Agbohessou et al., 2023; Bigaignon et al., 2020; Delon et al., 2019). In the litter decomposition GENDEC sub-model, the soil C content is calculated from the total litter input provided by STEP while soil N is derived from the quantity of C using C/N ratios (Moorhead and Reynolds, 1991). Soil moisture, soil temperature, and biomass (*i.e.,* herbal aerial mass, herbaceous root mass, ligneous leaves mass and faecal matter by livestock) are used as input variables to simulate microbial respiration. This is done by examining the interaction between buried litter, decomposer microorganisms and six C and N pools (*i.e.,* labile compounds, holo-cellulose, resistant compounds, dead microbial biomass, living microbial biomass and soil N). N$_2$O production and emissions from nitrification and denitrification are simulated using DNDC's (DeNitrification-DeComposition) equations (Li et al., 2000; Liu, 1996), adapted to the semi-arid region as described in Bigaignon et al. (2020) and Agbohessou et al. (2023). STEP alone has already been run to simulate aboveground biomass production, at the local scale (Jarlan et al., 2008, 2005, 2003; Mougin et al., 1995), meso scale (Grippa et al., 2017) and West-African Sahel scale (Pierre et al., 2016). Find in the appendices, the summary figure (Fig. A9) showing the connection between the models STEP, GENDEC, and the N$_2$O module.

### 2.2.2 Model up-scaling

We used STEP-GENDEC-N$_2$O to simulate daily herbaceous vegetation mass, CO$_2$ and N$_2$O emissions from soil in West Sahelian SPSs. We developed a framework to run the model at a regional scale, using the parameterizations developed in the above-cited studies. Simulations were performed at the West Sahelian band scale (Fig. 1) divided into 18271 grid cells of 0.1° x 0.1°, from 2012 to 2022. Input variables were extracted from different datasets available at the global or regional scale as described below (Table 1). For the soil dataset which is provided at finer resolution (<0.1° x 0.1°), pixel values for each centroid of the 18271 simulation grid cells were extracted. Simulations were performed over an 11-year period (2012-2022) preceded by a 6-year spin-up using the meteorological forcing data of year 2012, which was repeated 6 times. The spin up period allows for carbon and nitrogen pools to reach stability, as in Agbohessou et al. (2023). Indeed, in the model, the carbon

compartments for buried litter, feces, and dry roots are not initialized at 0, thus our simulations start with initial carbon values of 3.7, 0.3, and 6.0 gC for buried litter, feces, and dry roots, respectively. These values represent means derived from in situ measurements collected over several years at the Dahra site, where the model has been previously employed at the local scale. The carbon and nitrogen submodel used is relatively simple, employing first-order differential equations with moderate nonlinearity, which likely accounts for the rapid convergence observed in the model. All this explains why extensive spin-up time is not necessary for running the model with appropriately supplied carbon and nitrogen compartments.

**Table 1:**

**Summary of the datasets used for input variables and land cover/use**

| Dataset | Input variable (unit) | Spatial resolution | Temporal resolution | Ref | URL |
|---|---|---|---|---|---|
| **iSDAsoil** | Soil pH (-) and soil texture (clay, silt, and sand content (%)) | 30 m | 01-01-2012 (taken as constant) | (Hengl et al., 2021) | https://developers.google.com/earth-engine/datasets/tags/isda?hl=en |
| **ERA5-Land** | Initial soil water content (mm) and initial soil temperature (∘C) | 0.1° x 0.1° | 01-01-2012 (taken as constant) | (Muñoz Sabater, 2019) | https://cds.climate.copernicus.eu/cds app#!/dataset/reanalysis-era5-land?tab=form |
| **GPM_3IMERGDF** | Precipitation (mm) | 0.1° x 0.1° | 01-01-2012 to 31-12-2021 (daily) | (Huffman et al., 2019) | https://disc.gsfc.nasa.gov/datasets/GPM_3IMERGDF_06/summary?keywords=%22IMERG%20final%22 |
| **AgERA5** | Temperature (°C), Solar radiation (MJ m$^{-2}$), Vapour pressure (hPa), Wind speed (m s$^{-1}$), Soil albedo (-) | 0.1° x 0.1° | 01-01-2012 to 31-12-2021 (daily) | (Boogaard et al., 2020) | https://cds.climate.copernicus.eu/cds app#!/dataset/sis-agrometeorological-indicators?tab=overview |
| **Gridded Livestock of the World version 3 (GLW3)** | Animal load (-) | 0.083333 decimal degrees | 2012 (taken as constant) | (Gilbert et al., 2018) | https://dataverse.harvard.edu/dataverse/gld |
| **Action Contre la Faim Surveillance West Africa** | Proxy of herbaceous mass at germination (Kg ha$^{-1}$) | 1 km | 2019-2021 (taken as constant) | (Lambert et al., 2019; Bernard and Fillol, 2020, 2021) | https://data.humdata.org/dataset/acf_biomass_west-africa_raster |
| **Tree area density** | Proxy of trees' foliar biomass | 100 m | 2023 (taken as constant) | (Tucker et al., 2023) | https://daac.ornl.gov/cgi-bin/dsviewer.pl?ds_id=2117 |
| **Land Cove/Use product** | | | | | |
| **Global Land Cover – SHARE (GLC – SHARE)** | Land cover and land use (-) | 1-km | 2013 (taken as constant) | (Latham et al., 2014) | https://data.apps.fao.org/map/catalog/srv/eng/catalog.search#/metadata/ba4526fd-cdbf-4028-a1bd-5a559c4bff38 |

## 2.3 Model input data

### 2.3.1 Climate data

The climate data required for the simulation were derived from two different datasets (GPM_3IMERGDF and AgERA5). Precipitation (mm) data were taken from the IMERG (Integrated Multi-satellitE Retrievals for GPM) dataset, GPM_3IMERGDF (Huffman et al., 2019). GPM_3IMERGDF or GPM IMERG Final Precipitation L3 1 day 0.1 degree x 0.1 degree V06, is derived from the half-hourly GPM_3IMERGHH dataset (Huffman et al., 2019) and represents the final estimate of the daily accumulated precipitation. The selected product is "*precipitationCal\**: multi-satellite precipitation estimates with
gauge calibration". Dezfuli et al. (2017) validated the IMERG product in Africa using gauge data from West and East Africa. They showed that the precipitation diurnal cycle is relatively better captured by IMERG than by the TMPA (TRMM Multi-Satellite Precipitation Analysis) product. Maranan et al., (2020) did a process-based validation of GPM IMERG in Africa using gauge data from a West African forested zone. Additionally, the choice of the IMERG dataset over the ERA5 dataset for precipitation is based on expert recommendations and the results of previous evaluations of ERA5 precipitation data by Lavers
et al. (2022). Their study highlighted significant errors primarily in tropical regions. According to Lavers et al. (2022), users can only have confidence in ERA5 precipitation data in extratropical regions.

The spatial distribution of the GPM_3IMERGDF average precipitation over the last decade (2012-2022) exhibits significant gradients, with precipitation reaching as low as 0 mm at the northern border, exceeding 500 mm at the south-eastern border, and exceeding 1000 mm at the south-western borders (Fig. A2). Additionally, there is a significant increasing trend in annual
mean precipitation amounts from 2010 to 2021, along with interannual variability (Fig. 3c).

Temperature (°C), solar radiation (MJ m$^{-2}$), vapour pressure (hPa) and wind speed (m s$^{-1}$), were extracted from the AgERA5 dataset (Boogaard et al., 2020) using the R package "ag5Tools" (Brown and de Sousa, 2022). AgERA5 dataset provides daily surface meteorological data matching the input needs of STEP-GENDEC-N$_2$O. The dataset is actually based on the ECMWF (European Centre for Medium-Range Weather Forecasts) re-analysis ERA5-Land dataset (Muñoz Sabater, 2019). ERA5-Land
is an enhanced global dataset for the land component of the fifth-generation reanalysis produced by the ECMWF. It combines extensive historical observations from satellites, aircraft, land and marine weather sensors into global estimates using advanced modelling and data assimilation systems to generate consistent time series of multiple climate variables. More information about ERA5-Land product can be found in Muñoz Sabater et al. (2021) and Gleixner et al. (2020). In the data used, no significant trend (p>0.01) was observed in average air temperature (range: 25-35°C), minimum air temperature (range: 16-
27°C), maximum air temperature (range: 25-39°C), global radiation (range: 19-25 MJ m$^{-2}$ d$^{-1}$), wind speed (range: 2-7 m s$^{-1}$) and vapour pressure (range: 5-25 hPa) (extracted from ERA5-Land) in the Sahel between 2012 and 2022.

### 2.3.2 Soil data

Soil pH and soil texture (i.e. clay, silt, and sand content) were obtained from the iSDA (Innovative Solutions for Decision Agriculture Ltd.) soil dataset (Hengl et al., 2021). The iSDAsoil dataset contains soil property predictions at 30 m pixel size using machine learning coupled with remote sensing data and a training set of over 100,000 analysed soil samples all over Africa (Hengl et al., 2021; Miller et al., 2021). Prediction uncertainty estimates per pixel for the iSDA soil properties data are provided in Hengl et al. (2021). In the same study, the average accuracy performance based on fivefold spatial cross-validation for various soil variables indicated that soil pH exhibited the highest performance with a concordance correlation coefficient (CCC) of 0.90. The CCC values for soil clay content, sand content, and silt content were 0.85, 0.85, and 0.78, respectively. We initialized the dry soil albedo, soil moisture (mm) and soil temperature (℃) at the beginning of the simulation using data extracted from the ECMWF re-Analysis ERA5_Land (Muñoz Sabater, 2019).

Exploration of the extracted soil datasets showed that the soils in the Sahel region are typically sandy, with high levels of sand and low levels of clay (Fig. A3a and A3b). This results in well-drained soils but low in nutrients. The soil pH in the south-western part of the Sahel ranges from 5 to 7, while in the north and east it is higher than 7 (Fig. A3c). The pH levels of the soils in the Sahel vary also depending on their texture. Sandier soils typically have a high pH (7-8.5), while clay soils have a lower pH (5-7).

### 2.3.3 Animal load data

Information about livestock population and animal load distribution were obtained from the total livestock number for the reference year 2010 provided by the Gridded Livestock of the World version 3 (GLW3) (Gilbert et al., 2018) dataset. GLW3 provides global population densities of cattle, buffaloes, horses, sheep and goats in each land pixel at a spatial resolution of 0.083333 decimal degrees (approximately 10 km at the equator). The relative spatial distribution of livestock over the simulation period was assumed to be the same as the one indicated by the GLW3 database for the year 2010. To our knowledge no measurement data are available on the temporal variation of livestock across the Sahel. Indeed, FAOSTAT provides estimates of the livestock population at the national level for the period from 2012 to 2020 (FAOSTAT, 2024). However, these data are only available at the national scale and have not been downscaled to the finer spatial scales required for our simulation. GLW3 is currently the most recently compiled and harmonized subnational livestock distribution data available (and only covers the year 2010). In Gilbert et al. (2018) it is mentioned that the outputs of the GLW3 dataset have been adjusted to ensure that the total number of animals in a country aligns with the FAOSTAT 2010 total stock. There are no recent datasets available prior to 2010 presenting livestock distribution at the subnational scale in our region to our knowledge. We used the annual values of the GLW3 database to distribute the animal load on a monthly basis, taking into account the temporal variation of the livestock population from one month to the next throughout the year. We assumed an increase of the livestock up to 100% (in reference to the GLW3 database) in the pixels during the rainy season, and a gradual decrease down to 20% as we approach the middle of the dry season.

Analysis of the GLW3 dataset revealed that livestock is heterogeneously distributed across the Sahel and the animal load is dominated by bovines, ovines, caprines and some equines (Gilbert et al., 2018). High livestock densities were observed in north-western Senegal, southern Mauritania, central Mali, northern Burkina-Faso, southern Niger, northern Nigeria and south-western Chad (Fig. A3f).

### 2.3.4 Initial biomass data

The model calibration input parameters related to herbaceous vegetation such as Initial mass ($B_g0$) and Initial Specific Leaf Area ($SLA_g0$) at germination date were computed using data from the biomass dataset provided by "Action Contre la Faim (ACF) Surveillance West Africa" (Bernard and Fillol, 2021, 2020; Lambert et al., 2019). ACF biomass data were produced from 10-day images of Dry Mass Production (DMP) from SPOT-VEGETATION 4&5, PROBA-V and SENTINEL-3 satellites (Lambert et al., 2019). The retrieval algorithm of the DMP product is described as follows (Monteith, 1972; Swinnen et al., 2022) Eq. (1):

$$DMP = R \bullet fAPAR \bullet \mathcal{E}_{LUEc} \bullet \mathcal{E}_c \bullet \mathcal{E}_T \bullet \mathcal{E}_{CO2} \bullet CUE \tag{1}$$

$DMP$ is the 10-day Dry Matter Production (kgDM/ha/day), $R$ is the 10-day total shortwave incoming radiation ($GJ_T$/ha/day), $fAPAR$ is the PAR-fraction absorbed by green vegetation ($J_{AP}/J_P$), $\mathcal{E}_{LUEc}$ is the light use efficiency at optimum (kgDM/$GJ_{AP}$), $\mathcal{E}_c$ is the fraction of PAR in total shortwave ($J_P/J_T$), $\mathcal{E}_T$ is the normalized temperature effect, $\mathcal{E}_{CO2}$ is the normalized $CO_2$ fertilization effect and $CUE$ is the carbon use efficiency.

The 1 km$^2$ resolution biomass raster product showing biomass production in the Sahel in kg ha$^{-1}$ yr$^{-1}$ was downloaded for the study period. We extracted the biomass value for each centroid of the simulation grid cells and performed a normalization by linearly scaling the dataset to a range between 0 and 2.5 g m$^{-2}$ (the min and max values of $B_g0$ in the STEP model) to get the spatial distribution of the initial biomass ($B_g0$) at germination date. To obtain the spatial distribution of the initial Specific Leaf Area ($SLA_g0$) at germination date, we normalized ACF biomass dataset to a range between 0 and 280 cm$^2$ g$^{-1}$ (min and max values of $SLA_g0$ given in Jarlan et al. (2008)). Here is the normalization formula used to linearly scale biomass values to $B_g0$ and $SLA_g0$ ranges Eq. (2):

$$X_{norm} = a + \frac{(x-(x)) \bullet (b-a)}{(x) - min(x)} \tag{2}$$

With $X_{norm}$ representing the value of $B_g0$ or $SLA_g0$, $a$ and $b$ being the smallest and the largest value that $B_g0$ or $SLA_g0$ can take, respectively, and $x$ being the biomass values in the ACF dataset.

In the model $B_g0$ and $SLA_g0$ are calibration parameters. $B_g0$ mainly affects the date of peak biomass (Tracol et al., 2006), whereas $SLA_g0$ is used to estimate LAI (and fAPAR). The maximum conversion efficiency ($\mathcal{E}_c$) of absorbed radiation into biomass (i.e., g of dry matter per MJ of absorbed photosynthetically active radiation) was set to 5 g MJ$^{-1}$ which corresponds to the central value of the $\mathcal{E}_c$ range possible values (Mougin et al., 1995; Pierre et al., 2011; Tracol et al., 2006) for all simulation grid cells.

### 2.3.5 Foliar mass of trees

Using the allometric equation developed by Hiernaux et al. (2023), we transformed the tree area density product provided by Tucker et al. (2023) into an estimate of tree foliar biomass in each simulation grid cell (Fig A3e). The conversion formula employed was Eq. (3):

$$DM_{foliar} = 0.2693 \bullet A^{0.9441} \tag{3}$$

Here, $DM_{foliar}$ represents the mass of trees' leaves in kilograms, and $A$ denotes the tree crown area in square meters.

### 2.4 Accounting for SPSs distribution in model outputs

Global Land Cover – SHARE (GLC – SHARE) dataset (Latham et al., 2014) provides information about the spatial distribution of a set of eleven major land cover classes (i.e., artificial surfaces, cropland, grassland, tree covered areas, shrubs covered areas, herbaceous vegetation, aquatic or regularly flooded, mangroves, sparse vegetation, bare soil, snow and glaciers, and water bodies) for the year 2013 and at 1-km$^2$ pixel resolution. First, we assumed that land cover change intensity was negligible in the Sahel during the last decade (the study period). Second, a new land cover class called silvopastoral areas was created and represents the sum of pixels of the classes: shrubs covered areas and grassland (Fig. 1).

The proportions of silvopastoral areas pixels within the $0.1° \times 0.1°$ simulation grid cells (pixel resolution $\approx 123.21$ km$^2$) were calculated using the GLC – SHARE dataset to obtain the spatial distribution of silvopastoral systems in the Sahel (Fig. A1). In our analysis and interpretation of the spatial distribution of herbaceous mass, $CO_2$, and $N_2O$ emissions, we consider the model outputs for simulation pixels where silvopastoral areas are > 80%. Additionally, bivariate maps were proposed, which display both model outputs and the distribution of SPSs in the simulation domain, to provide a more comprehensive view of the results.

To estimate the annual budget of soil $CO_2$ and $N_2O$ emissions, the model outputs were weighted by the proportion of silvopastoral area within each simulation grid cell (Fig. 1 and Fig. A1), therefore considering all SPSs, even those which %SPS < 80%, across the simulation domain.

### 2.5 Random Forest algorithm for the analysis of soil CO₂ and N₂O emissions driving parameters

Random Forest (RF) is a machine learning method developed by Breiman (2001), it is a natural non-linear modelling tool that has proven valuable in many fields (Liu et al., 2022; Webb et al., 2021). We used the RF algorithm to identify the most important factors influencing the spatial distribution of soil $CO_2$ and $N_2O$ emissions. The main advantages of RF algorithms are its low number of tunable factors, good tolerance to outliers and noise, general resistance to overfitting, and ability to identify and rank the most important variables (Liu et al., 2022; Webb et al., 2021). The RF algorithm was implemented into the R software (R Core Team, 2019) and the modelling framework provided by the *randomForest* R package (Liaw and Wiener, 2002) is used in our study. The target variables of the RF are the spatial distribution of the simulated soil $CO_2$ and soil

N$_2$O emissions, while the explanatory variables include the spatial distribution of various environmental and biological factors susceptible to impact the spatial distribution of the soil CO$_2$ and N$_2$O emissions simulated by the STEP-GENDEC-N$_2$O model. These factors consist of a combination of output variables from the STEP-GENDEC-N$_2$O model (e.g., soil water content, soil temperature, soil C content, soil N content and herbaceous mass) and input variables for the STEP-GENDEC-N$_2$O model (e.g., soil sand content, soil clay content, soil pH, air temperature, albedo, annual precipitation and animal load). We conducted the RF with the default parameters proposed by the *randomForest* package.

The method is composed of three critical steps, each of which plays a crucial role in the overall performance of the model. In the first step, a bootstrap sample of observations (equal to the number of trees) is randomly drawn from the dataset, with replacement. Approximately one third of the total observations are left out and used as "out-of-bag" (OOB) data to evaluate the model's performance and prevent the need for a separate validation dataset (Efron and Tibshirani, 1986; Philibert et al., 2013). This provides a resampling procedure that generates multiple versions of the training dataset, which helps to mitigate overfitting and improves the accuracy of the model. In the second step, a random subset of predictor variables is selected at each node of the decision tree (Ghattas, 2000; Philibert et al., 2013; Prasad et al., 2006). The number of variables selected (mtry) was set to the integer part of the square root of the total number of variables (Breiman, 2001; Liaw and Wiener, 2002; Philibert et al., 2013). This approach involves considering a subset of variables at each node of the decision tree and selecting the best variable that maximizes the information gain. This randomization technique reduces the correlation among the trees and makes the model more robust and accurate. In the final step, multiple decision trees are grown from the bootstrapped dataset and the random subsets of features. The trees are grown using recursive binary partitioning of the data, with the best split determined by optimizing a quality criterion such as information gain by the Gini impurity index (Breiman et al., 1984). The final prediction is made by aggregating the predictions of all trees in the forest by averaging the outputs. The process is repeated multiple times until a stable estimate of model performance is obtained.

We assessed variable importance using the percentage increase in Mean Squared Error (*%IncMSE*) after a factor was randomly permuted. *%IncMSE* estimates the contribution of each variable to the reduction in the mean squared error of the model (Breiman, 2001; Echeverry-Galvis et al., 2014). Factors with higher *%IncMSE* values are considered as more important in explaining the spatial distribution of soil CO$_2$ and N$_2$O emissions. The importance of each factor was displayed with the variable importance plot developed from the RF.

### 2.6 Statistical analysis and mapping

We conducted linear regression analysis to examine trends over time in herbaceous vegetation mass, soil CO$_2$ and N$_2$O emissions and relevant emission driving variables. The Pearson correlation was used to assess the relationship between the different variables. All statistical analysis and mapping were performed using R (R Core Team, 2019).

# 3 Results

## 3.1 Spatio-temporal patterns of aboveground herbaceous mass in the Sahelian SPSs (2012-2022)

The annual production of aboveground herbaceous mass in the Sahelian SPSs, simulated from 2012 to 2022, displays a latitudinal gradient characterized by higher herbaceous mass in the southern regions, which diminishes as we progress towards the northern latitudes (Fig. 2). The same spatial pattern is observed in Figure 2b; which highlights results for Sahelian SPSs (pixel %SPS>80%). The maximum annual mean production (2012-2022) reaches 3 t DM ha$^{-1}$ yr$^{-1}$ and the annual minimum production is 0 t DM ha$^{-1}$ yr$^{-1}$.

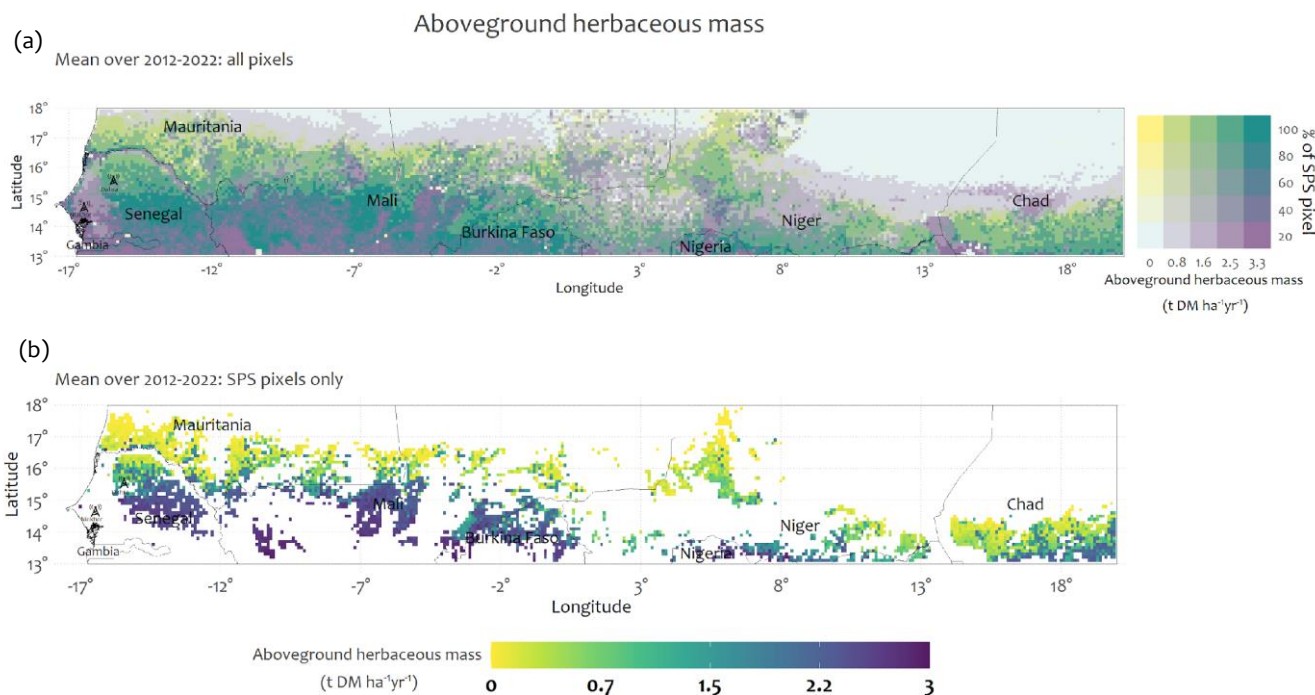

**Figure 2: Regional distribution of simulated herbaceous mass in the Sahelian SPSs (annual mean over 2012-2022), in tDM ha-1 yr-1. (a) Bivariate map, which displays both simulated herbaceous mass and the distribution of SPSs in the simulation domain, (b) map filtering the simulated herbaceous mass for areas with Sahelian SPSs >80% only.**

Herbaceous mass in Sahelian SPSs exhibited inter-annual variations with standard deviations reaching up to 1.3 t DM ha$^{-1}$ yr$^{-1}$ at some locations (Fig. A4a). We observed a significant increasing trend (p<0.001) in the annual herbaceous mass anomaly (deviation from the 2012-2022 average) from 2012 to 2020 (Fig. 3a). This rising trend is evident in the Hovmöller representation, which depicts a gradual increase in herbaceous mass, particularly in the southern Sahel region around the latitudes 13°N and 15°N (Fig. 3b), with the highest production simulated in the wettest years (2019, 2020 and 2021, Fig. 3c). In the southern Sahel (13°N to 15°N), herbaceous mass in SPSs can reach 2.5 t DM ha$^{-1}$ yr$^{-1}$, while in the northern Sahel (16°N

to 18°N), it does not exceed 0.5 t DM ha$^{-1}$ yr$^{-1}$ (Fig. 3b). Overall, herbaceous mass in the Sahelian SPSs is highly correlated to the wet season total precipitation which shows large inter-annual variation (Fig. 3c: p < 0.001 and r = 0.6).

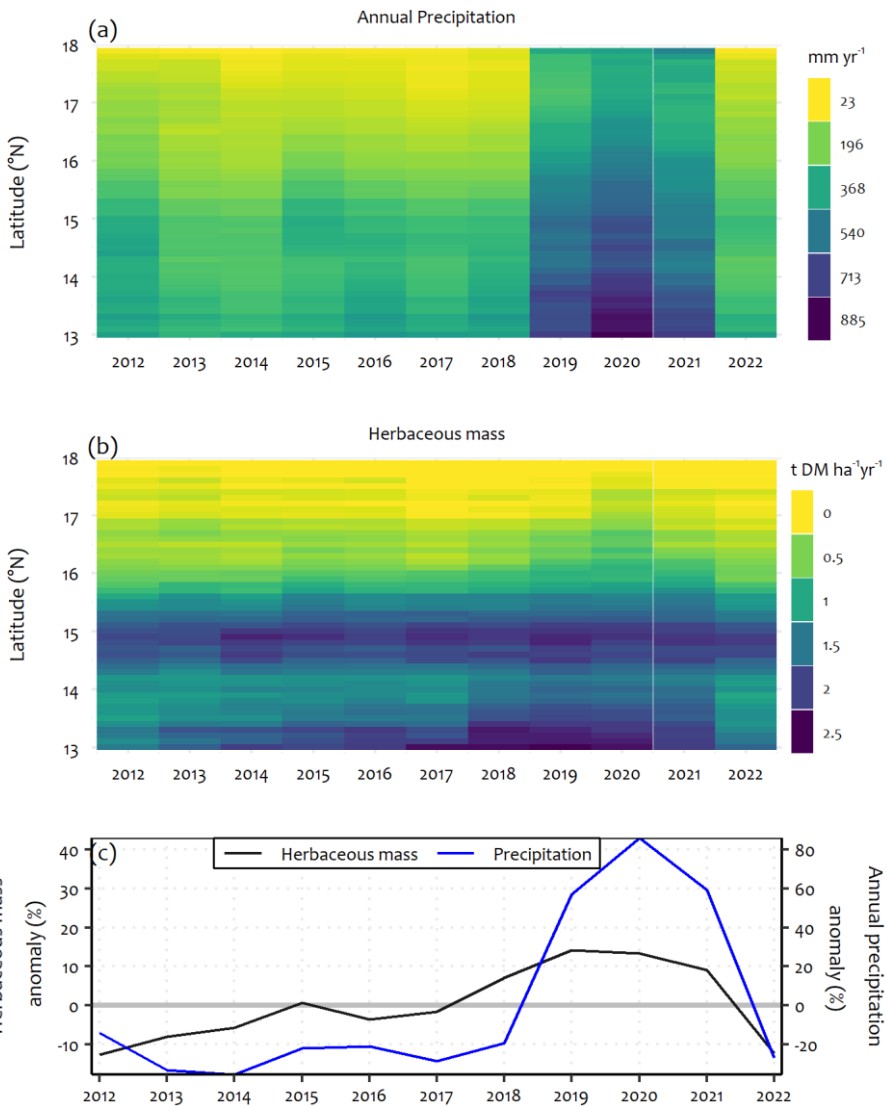

Figure 3: (a) Hovmöller (latitude-year) plot of the annual precipitation. (b) Hovmöller (latitude-year) plot of herbaceous mass in the domain indicated in Fig. 2b. (c) Interannual variations of anomalies (relative to the mean value for the period 2012-2022).

**3.2 Soil CO$_2$ and N$_2$O emissions in Sahelian SPSs**

**3.2.1 Spatial distribution across the Sahel**

The simulation results reveal a heterogeneous spatial distribution of soil CO$_2$ and N$_2$O emissions, with the lowest emissions in the north and the highest emissions in the south (Fig. 4). SPSs in the pastoral zones of central Senegal, in southern and central Mali, in northern Burkina Faso, and in southern Niger (between longitudes 7°E and 8°E) exhibit high levels of soil CO$_2$ emissions (Fig. 4a and 4b). The average soil CO$_2$ emissions for the period 2012-2022, reached up to 1.7 t CO$_2$-C ha$^{-1}$ yr$^{-1}$, as shown in figure 4b. SPSs located in the northern regions of Niger, as well as in Mauritania, were generally not significant sources of CO$_2$ (Fig. 4b). Only SPSs of central Senegal, northern Burkina Faso and Mali remain constant CO$_2$ emission hotspots throughout the study period, with emissions as high as 2.6 t CO$_2$-C ha$^{-1}$ yr$^{-1}$ in some years, as shown in the all-years detailed maps in figure A6. Inter-annual variabilities of up to 0.7 t CO$_2$-C yr$^{-1}$ ha$^{-1}$ have been observed in some SPSs (Fig. A4b).

Figure 4c depicts heterogeneous soil N$_2$O emissions ranging from 0 to 3 kg N$_2$O-N ha$^{-1}$ yr$^{-1}$ and high emissions in some areas where the percentage of SPSs pixel is lower than 80%. The figure 4d exclusively show cases areas that are representative of the Sahelian SPSs (%SPS>80), showing that soil N$_2$O emissions were as high as 2.3 kg N$_2$O-N ha$^{-1}$ yr$^{-1}$ (mean 2012-2022 period) in SPSs located within the sandy pastoral zones of central Senegal, and in southern Mali between latitudes 13°N and 15N°E. In contrast, smaller N$_2$O emissions were observed in the other SPSs of the region, especially in Niger and Chad. High inter-annual variabilities have been observed in the southern part of the Sahel (Fig. A4c).

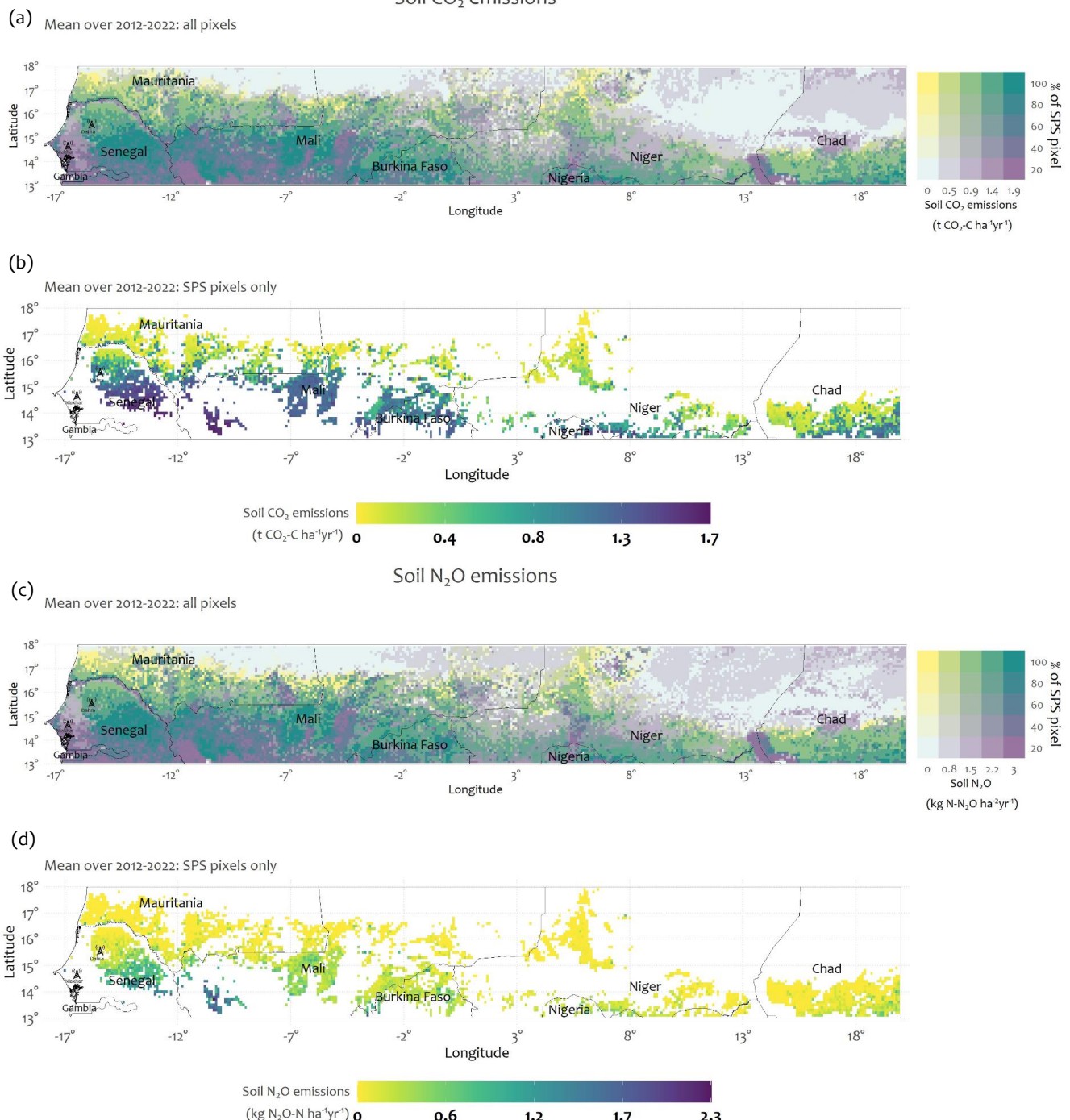

Figure 4: Regional distribution of simulated soil CO₂ andN₂O emissions in the Sahelian SPSs (annual means over 2012-2022). (a) and (c): Bivariate maps display both model outputs and the distribution of SPSs in the simulation domain. (b) and (d): Maps displaying model outputs only on areas representative of the Sahelian SPS (>80%).

### 3.2.2 Exploring the temporal dynamics of model outputs

Figure 5 shows the temporal dynamics of wet season precipitation, soil $CO_2$ emissions, soil $N_2O$ emissions, soil water content and soil total C at two contrasted sites showing different emission levels (low and high), located in Niger (longitude 10.7, latitude 14.2) and Senegal (longitude -15.4, latitude 15.4) respectively. These sites were on predominant sandy soils. The observed dynamics of the different variables (precipitation, soil $CO_2$ emissions, soil $N_2O$ emissions, soil water content and soil C content) at these sites show the model's ability to simulate realistically seasonal variations at fine time-scales in soil $CO_2$ and soil $N_2O$ emissions in the Sahel.

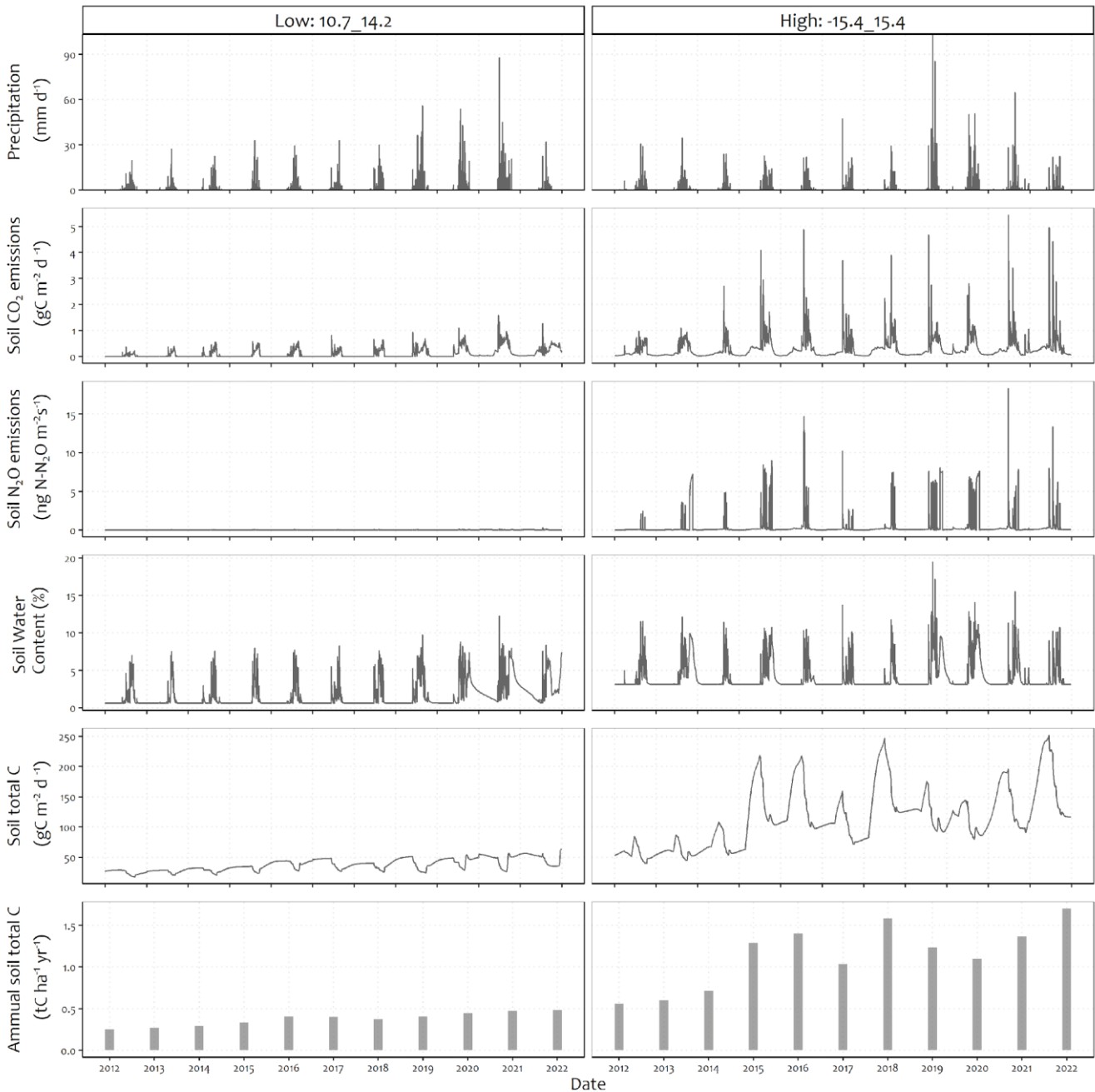


**Figure 5: Temporal dynamics of model outputs across two sites with different levels of soil $CO_2$ &$N_2O$ emissions. From top to bottom: precipitation, soil $CO_2$ emissions, soil $N_2O$ emissions, soil water content and soil total C. At the left, a site exhibiting low emissions (latitude -10.7, longitude 14.2), at the right, a site with high emissions (longitude -15.4, latitude 15.4).**

### 3.2.3 Factors controlling the spatial distribution of soil $CO_2$ and $N_2O$ emissions

The observed variations in the spatial patterns of soil $CO_2$ and $N_2O$ emissions were attributed to a complex interaction between meteorological, edaphic, bio-physical factors. According to a statistical analysis assessed by Random Forest over the model output in grid cells containing more than 80% of SPSs, the soil carbon and nitrogen contents were found to be the primary factors controlling the spatial distribution of soil $CO_2$ and $N_2O$ emissions, respectively, as shown in figure 6. Soil C content, air temperature and soil temperature were identified as the three most significant factors controlling the spatial patterns of soil

$CO_2$ emissions. For soil $N_2O$, the two most significant factors after soil N content were soil water content and animal load. The results further showed that for soil $CO_2$, the other driving factors were herbaceous mass, animal load, annual precipitation (or soil water content), soil clay content, and soil water content (Fig. 6a). For soil $N_2O$, herbaceous mass, soil temperature, soil clay content, annual precipitation (or soil water content), and air temperature (in that order) also appeared as key driving factors (Fig. 6b). Soil pH was found to have the least influence on the spatial pattern of soil $N_2O$ emissions (Fig 6).

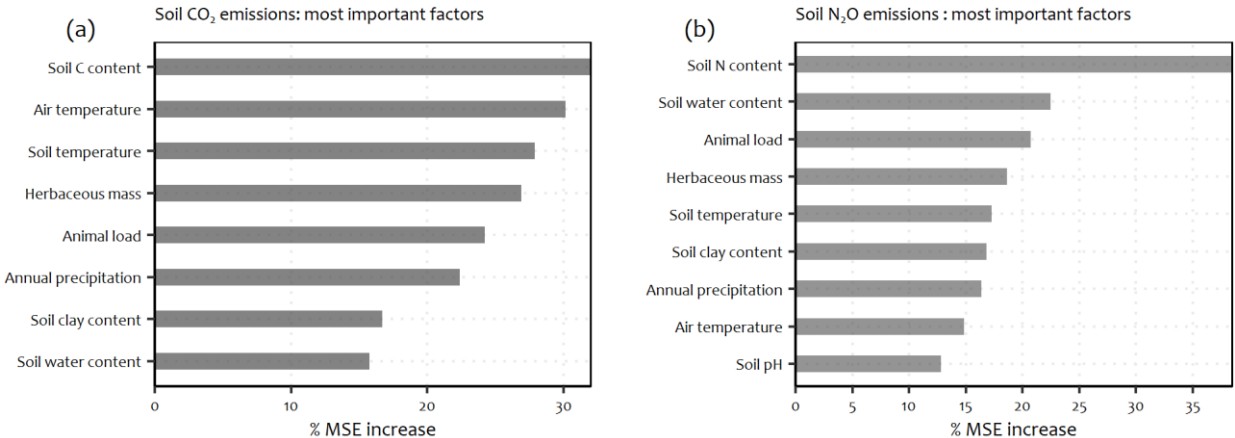


**Figure 6: Factors controlling the spatial changes in (a) soil $CO_2$ and (b) $N_2O$ emissions, from Random Forest Analysis. MSE = Mean Squared Error**

### 3.2.4 Annual budgets across the Sahel (2012-2022)

The simulated soil $CO_2$ emissions include both microbial respiration and root respiration of herbaceous. Between 2012 and

2022, the estimated average soil $CO_2$ emissions in the Sahelian SPSs was 58.79 ±4.83 Tg $CO_2$-C $yr^{-1}$ (1 Tg = $10^{12}$ g). The highest soil $CO_2$ annual emission (65.80 Tg $CO_2$-C $yr^{-1}$) was found in 2020, while the lowest (50.77 Tg $CO_2$-C $yr^{-1}$) was in 2012 (Fig. 7a). During this same period, the mean soil $N_2O$ emission was 21.59 ± 3.91 Gg $N_2O$-N $yr^{-1}$ (1 Gg = $10^9$ g), ranging from 17.31 Gg $N_2O$-N $yr^{-1}$ in 2012 to 27.43 Gg $N_2O$-N $yr^{-1}$ in 2020 (Fig. 7b). From 2012 to 2020, annual soil $CO_2$ and $N_2O$ emissions showed significant ($p < 0.01$) rising trends of 4.30e-3 ± 6.05e-4 Tg $CO_2$-C $yr^{-1}$ and 3.75e-3 ± 4.47e-4 Gg $N_2O$-N $yr^{-1}$

$^1$, respectively. However, emissions dropped after 2021, with a 17.5% decrease in soil $CO_2$ emissions and 25.5% decrease in soil $N_2O$ emissions (Fig. 7c). Figure 7c reveals that the inter-annual variations in soil $CO_2$ and soil $N_2O$ emissions are quite

homothetic, as indicated by a Pearson's correlation coefficient of 0.86. Annual precipitation over the 2012-2022 period and averaged over the study domain was significantly correlated to both soil $CO_2$ ($p<0.05$, $r=0.48$) and $N_2O$ ($p<0.05$, $r = 0.79$) emissions.

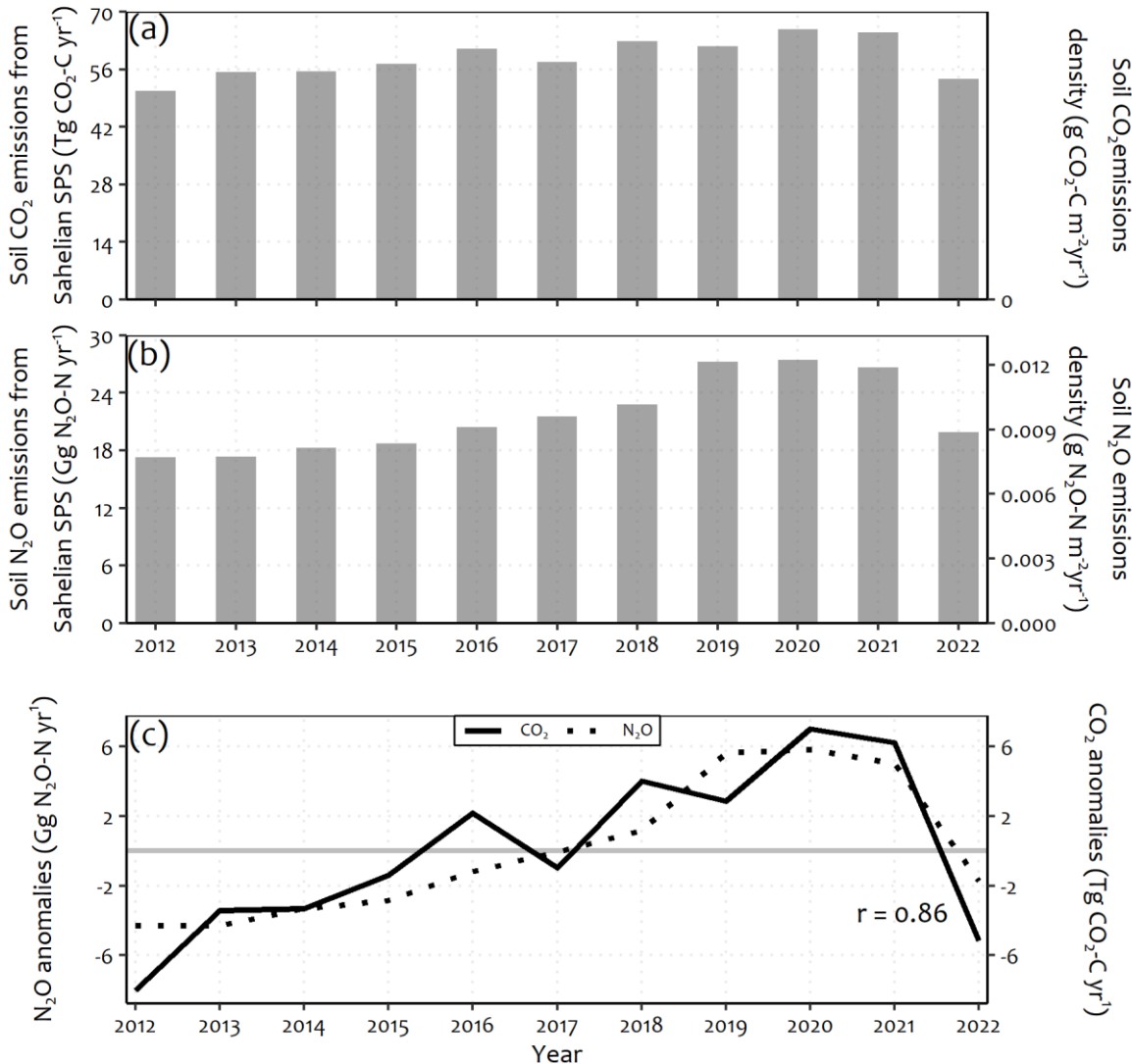


**Figure 7: Interannual variation in soil $CO_2$ and $N_2O$ emissions in the Sahelian SPSs (which covers *approx.* 892000 km² ) during 2012-2022. (a) Soil $CO_2$ emissions in Tg C yr⁻¹ (1 Tg = 10¹² g) and (b) soil N2O emissions in Gg C yr⁻¹ (1 Gg = 10⁹ g). (c) Interannual variations of soil $CO_2$ and soil $N_2O$ anomalies (relative to the mean value for the period 2012-2022). The Pearson correlation coefficient between $CO_2$ and $N_2O$ anomalies was 0.86. We calculated the proportion of SPSs area pixels within each 0.1° x 0.1°**

**simulation grid cell, and used these values to weight the model outputs for each grid cell.**

## 4 Discussion

Previous studies at global and regional scales have estimated greenhouse gas (GHG) emissions from various ecosystems, especially agricultural systems (Tian et al., 2020, 2015), forests (Tian et al., 2020; Verchot et al., 1999), and rangelands (Dangal et al., 2020). These studies have frequently highlighted significant uncertainties when estimating emissions from underrepresented regions, like in Africa. In addition, different modeling techniques often give divergent results when estimating emissions from these regions. In this study, we have up-scaled the 1D STEP-GENDEC-$N_2O$ model, which was previously used in local studies across various sites in the West Sahel region. For example, in previous studies conducted at a SSP located in the northern region of Senegal (Dahra: 15°24′10″N, 15°25′56″W), Bigaignon et al. (2020) effectively used STEP-GENDEC-$N_2O$ to satisfactorily simulate soil water content ($R^2 = 0.68$ & RMSE = 1.67 mm d$^{-1}$), $NO_3^-$ content in soil ($R^2 = 0.42$ & RMSE = 0.83 mgN kgsoil$^{-1}$), and $N_2O$ emissions ($R^2 = 0.36$ & RMSE = 2.51 ng N m$^{-2}$ s$^{-1}$). At the same site, Agbohessou et al. (2023) successfully simulated $CO_2$ fluxes using STEP-GENDEC-$N_2O$ combined with a tree growth model (DynACof (Vezy et al., 2020)), achieving convincing results for gross primary productivity (GPP: EF = 0.49 & RMSE = 2.15 gC m$^{-2}$ d$^{-1}$) and ecosystem respiration (Reco: EF = 0.56 & RMSE = 1.34 gC m$^{-2}$ d$^{-1}$). Additionally, Delon et al. (2019) demonstrated successful simulation of soil respiration on the identical site using STEP-GENDEC-$N_2O$. On another SSP located in Mali (Agoufou: 15.34° N, 1.48° W), Delon et al. (2015) employed STEP-GENDEC-$N_2O$ to satisfactorily simulate soil moisture ($R^2 = 0.7$), soil temperature ($R^2 = 0.86$), and herbaceous mass ($R^2 = 0.72$), yielding satisfactory results. Building upon these previous local applications and validations of the STEP-GENDEC-$N_2O$ model in different representative sites of the Sahelian SPS, we provide the first large-scale estimate of soil $CO_2$ and $N_2O$ emissions from W-Sahelian SPS.

In this section, we discuss the magnitude of soil $CO_2$ and $N_2O$ emissions reported in this study, the role of environmental and biological factors that drive the spatial heterogeneity observed in soil $CO_2$ and $N_2O$ emissions in Sahelian SPSs, and the uncertainties and limitations associated with these estimations.

### 4.1 Spatial and temporal patterns of herbaceous vegetation, soil CO2 emissions, and their relationship

In a previous study, Pierre et al. (2016) demonstrated the ability of the STEP model (alone) to simulate the dynamics of herbaceous vegetation at regional scale in the W-Sahel. They found a good agreement between the regional spatial patterns of STEP-simulated vegetation masses and the Moderate Resolution Imaging Spectroradiometer (MODIS) vegetation indices. They observed a latitudinal gradient in herbaceous vegetation mass, caused by the rainfall gradient, as also shown in our results. The magnitudes of herbaceous mass in their study and ours are comparable and the spatial patterns are similar although the study periods and the input data employed are not exactly the same. Previous estimates of mass production in the Sahel, using the LandscapeDNDC model (Rahimi et al., 2021) exhibited relatively stable temporal dynamics in mass production from 2010 to 2019. These estimates encompassed all land use types in the Sahel region, which could explain the divergence with our results showing a gradual increase in mass production in Sahelian SPSs between 2012 and 2022. Moreover, the trend observed in this study is mainly driven by the most recent years, with the highest values occurring in 2019, 2020 and 2021.

We compared the aboveground herbaceous mass (ABG) simulated by STEP-GENDEC-N$_2$O with the ABG biomass product

from ACF (Bernard and Fillol, 2020, 2021; Lambert et al., 2019) for SPS pixels only (Fig. 8). This revealed a significant

correlation between the ABG herbaceous mass simulated by STEP-GENDEC-N$_2$O and the ACF biomass product (with an $R^2$

value of 0.61 and an RMSE of 1.51). The ABG biomass derived from ACF amounts to 7 tDM ha$^{-1}$ yr$^{-1}$, whereas the simulated

ABG herbaceous mass by STEP-GENDEC-N$_2$O does not exceed 3 tDM ha$^{-1}$ yr$^{-1}$. This variation can be attributed to the ACF

product being derived from satellite data, encompassing not only herbaceous plants but also including the tree and crop

component within these SPS-dominated pixels. Additionally, the Monteith formulation (Monteith, 1972) used by ACF

approaches a potential biomass and therefore corresponds more to the upper bound of the STEP-GENDEC-N$_2$O simulations.

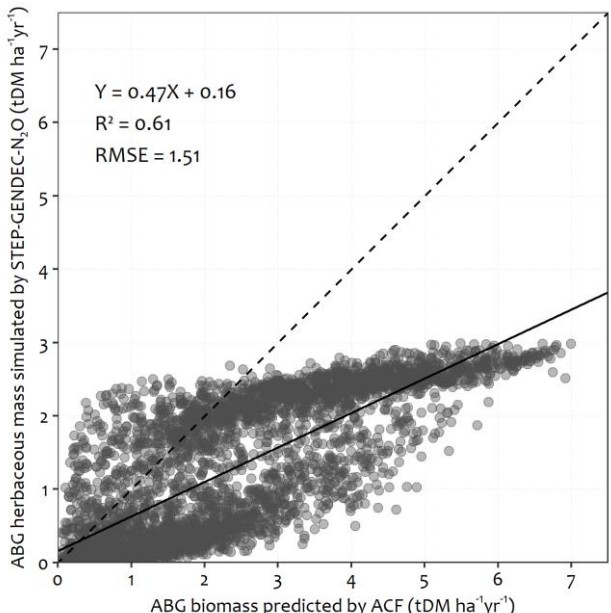

**Figure 8: The relationship between aboveground (ABG) herbaceous mass simulated by STEP-GENDEC-N2O and ABG biomass predicted by ACF (Bernard and Fillol, 2020, 2021; Lambert et al., 2019). Each point represents the annual mean biomass in a**
**simulation pixel. The dashed line represents the 1:1 line, while the solid line depicts the linear regression line.**

Plants' litter is the main source of carbon entering the soil, which explains the similar spatial patterns observed in both annual

herbaceous mass (Fig. 2b) and annual soil CO$_2$ emissions (Fig. 4b). This illustrates the effect of the C substrate on CO$_2$

emissions, as confirmed by the Random Forest analysis (Fig. 6). The size and composition (nature of substrate, molecules,

C/N ratio, etc.) of the available carbon pool actually control the magnitude of the CO$_2$ emissions from soil (Barnard et al.,

2020). Soil CO$_2$ emissions include the respiration of soil microorganisms (microbial or heterotrophic respiration) and plant

roots (autotrophic respiration), including all respiratory processes occurring in the rhizosphere (Raich and Potter, 1996; Xu

and Shang, 2016). Root cells perform cellular respiration, metabolizing carbohydrates that are sent down from the leaves.

Depending on the vegetation density, root respiration can contribute significantly to the total soil respiration (Macfadyen,

1970). In some SPSs in the north-western Sahel (*e.g.,* Mauritania, Mali and Niger), we simulated significant soil CO$_2$ emissions

despite the low herbaceous mass. These areas also exhibit high interannual variabilities in soil $CO_2$ emissions (Fig. A4b: up to 0.7 t $CO_2$-C ha$^{-1}$ yr$^{-1}$). The northern Sahel is generally characterized by a long dry season and very low rainfall. In such semi-arid areas, at the onset of the wet season, the first rainfall events rewet the dry soil resulting in a mineralization peak leading to a large soil $CO_2$ efflux pulse, also known as the "Birch effect" (Birch, 1958). The STEP-GENDEC-N$_2$O model accounts for this "Birch effect" (Delon et al., 2019), which could explain the soil $CO_2$ emissions hotspots simulated in some SPSs of the north-western Sahel. The site (simulation pixel) located at longitude -15.4°W and latitude 15.4°N (0.1 * 0.1 degrees), as depicted in Figure 5, actually illustrates the Birch effect in soil respiration dynamics, with notably high emissions simulated at the onset of the rainy seasons. This simulation pixel encompasses the Dahra site in northern Senegal (longitude -15.43222°W and latitude 15.40277°N), where the 1D STEP-GENDEC-N$_2$O model results were in good agreement with observations (Agbohessou et al., 2023; Delon et al., 2019). According to Fan et al. (2015), up to 20% of the annual soil $CO_2$ emissions to the atmosphere occurs in African savanna ecosystems following intense rainfall. The $CO_2$ pulses associated with rewetting can represent a large part of the annual C budget in semi-arid and arid ecosystems (Barnard et al., 2020; Jarvis et al., 2007; Ma et al., 2012; Rey et al., 2017).

In a SPSs located in northern Senegal, Delon et al. (2017) measured soil respiration ranging from 2.4 ± 0.62 gC m$^{-2}$ d$^{-1}$ at the onset of the wet season to 0.7 ± 0.01 gC m$^{-2}$ d$^{-1}$ at the end of the wet season in 2013. Our estimated mean soil $CO_2$ emissions density for Sahelian SPSs between 2012 and 2022 (0.06 gC m$^{-2}$ d$^{-1}$) is lower than estimates at the global scale for grasslands (2.2 gC m$^{-2}$ d$^{-1}$) and partially vegetated deserts (1.0 gC m$^{-2}$ d$^{-1}$) by Xu and Shang (2016). On a global scale, for these grasslands, the substrate (soil C content) is probably much more important than in SPSs, which explains the higher values of $CO_2$ emissions. Our simulated soil $CO_2$ emissions for our region are also lower than the estimates by Warner et al. (2019). The soil $CO_2$ emissions (soil respiration) calculated for our region (our simulation grid cells) from the Warner et al. (2019) product indicate values as high as 7.8 tC ha$^{-1}$ yr$^{-1}$, whereas the simulated soil $CO_2$ emissions by STEP-GENDEC-N$_2$O do not exceed 2 tC ha$^{-1}$ yr$^{-1}$. These differences can be explained by the following points: (i) The Warner et al. (2019) product is a one-time prediction based on input data from 01-01-1963 to 31-12-2011, while our simulated soil $CO_2$ emissions used for comparison represent the annual mean of the period from 2012 to 2022. (ii) We used a process-based model (STEP-GENDEC-N$_2$O), while the soil $CO_2$ emissions (soil respiration) predicted by Warner et al. (2019) are based on a machine learning approach, specifically a quantile regression forest model. This model was trained using selected environmental predictors and 2,657 input soil respiration observations from the global soil respiration database (SRDB) (Bond-Lamberty and Thomson, 2010). However, it's important to note that the SRDB database used by Warner et al. (2019) does not contain measurements from sites located in our region (the simulation area). Additionally, Warner et al. (2019) mentioned that the greatest prediction uncertainties were observed in semi-arid ecosystems.

## 4.2 Soil $N_2O$ and $CO_2$ emissions in Sahelian SPSs and importance of livestock

Between 2012 and 2022, the simulated soil $N_2O$ emissions from Sahelian SPSs were $0.022 \pm 0.004$ Tg $N_2O$-N yr$^{-1}$. The regional natural soil $N_2O$ emissions in Africa was estimated at 1.6 Tg $N_2O$-N yr$^{-1}$ for the period 2007-2016 (Tian et al., 2020). The simulated average soil $N_2O$ emissions from Sahelian SPSs is lower than the median total $N_2O$ emissions of 0.05 Tg $N_2O$-N yr$^{-1}$ from bomas (livestock enclosure where livestock excreta accumulate) in sub-Saharan Africa's semi-arid and arid climates (Butterbach-Bahl et al., 2020). The average soil $N_2O$ emission density (per unit area) in Sahelian SPSs (2012-2022) was found to be 0.01 g $N_2O$-N m$^{-2}$ yr$^{-1}$ (range: 0-0.23 g $N_2O$-N m$^{-2}$ yr$^{-1}$), which is comparatively lower than the average estimate in tropical regions ($0.11 \pm 0.02$ g $N_2O$-N m$^{-2}$ yr$^{-1}$) and the global average ($\approx 0.05$ g $N_2O$-N m$^{-2}$ yr$^{-1}$) reported for the period 2007-2016 (Tian et al., 2019). The soil $N_2O$ emission density in Sahelian SPSs (2012-2022) was also lower than global emission densities estimated in croplands ($0.21 \pm 0.08$ $N_2O$-N m$^{-2}$ yr$^{-1}$) and other ecosystems ($0.06 \pm 0.01$ g $N_2O$-N m$^{-2}$ yr$^{-1}$), respectively, during the period 2007-2016 (Tian et al., 2019). The most significant soil N input in Sahelian SPSs actually originates from livestock excreta, which is lower than the N input in most fertilized agricultural fields (Dangal et al., 2020), explaining the lower emission density in SPSs compared to the global average emissions density in croplands. In fact, studies have shown that nitrogen fertilizer application in croplands is the leading factor responsible for the increases in emission from agriculture (Cao et al., 2018; Davidson, 2009; Maavara et al., 2019; Shcherbak et al., 2014; Yao et al., 2020), followed by a minor yet significant rise in emissions from livestock manure (Tian et al., 2020). But on the other hand, in regions where very little nitrogen fertilizer is used in cropland such as in Africa, soil $N_2O$ emissions mainly arise from livestock manure deposited in pastures and rangelands (Butterbach-Bahl et al., 2020; Dangal et al., 2020; Xu et al., 2019). This confirms the $N_2O$ emission hotspots simulated in locations where the density of livestock is high (Fig. 4c & 4d, Fig A3f), as also highlighted by the Random Forest analysis. Indeed, the animal load distribution also affects the spatial distribution of soil $N_2O$ and $CO_2$ emissions, as shown in figure 6. Several authors have already mentioned this impact (Assouma et al., 2017; Butterbach-Bahl et al., 2020; Dangal et al., 2020; Smith et al., 2003). Livestock influences the spatial distribution of soil C and N, which, in turn, significantly affects soil $N_2O$ and $CO_2$ emissions.

## 4.3 Common features of soil $CO_2$ and $N_2O$ emissions in Sahelian SPSs

Figure 7c shows that the inter-annual variations in soil $CO_2$ and soil $N_2O$ emissions are quite homothetic, as indicated by a Pearson's correlation coefficient of 0.86. This suggests that they are both responding in a similar manner to the different ecological drivers. Some authors stated that the main processes responsible for $CO_2$ (decomposition) and $N_2O$ (nitrification and denitrification) emissions from soils are influenced by the same environmental factors, namely soil moisture, soil temperature, soil texture, soil C and N content (Davidson and Swank, 1986; Oertel et al., 2016; Rastogi et al., 2002; Signor and Cerri, 2013). Several studies have shown how soil $CO_2$ and $N_2O$ emissions evolve over time in response to changes in environmental driving factors (Cuhel et al., 2010; Davidson and Swank, 1986; Khalil, 2015; Ray et al., 2020) but the complexity of the interactions between these different factors make it difficult to assess the importance of each driver

responsible for the spatial distribution of the emissions. From our results, the main factor responsible for the spatial distribution of soil $CO_2$ and $N_2O$ emissions in SPSs (Fig. 6) is substrate availability (soil C and N content), which outweighs other factors such as soil water content, temperature, and soil texture. Moreover, substrate availability is directly linked to herbaceous mass productivity (as mentioned in section 4.1) and to animal load (see section 4.2). This is consistent with the findings of Ray et al. (2020), who showed that soil $CO_2$ emissions are affected more by substrate availability than by rainfall, although their experiment was performed in a cropping system. In addition to influencing the spatial pattern of soil $CO_2$ and $N_2O$ emissions, soil C and N also impact the temporal variation of these emissions, as shown in figure 5, where the largest emissions are found where the C content was the highest. Furthermore, our simulations revealed a rise in emissions between 2012 and 2020 (Fig. 7c), that is correlated to the increase in herbaceous mass during the same period (Fig. 3a). Indeed, the results produced by the Random Forest approach (Fig. 6) confirm our expectations regarding the soil C and N content as the primary factor influencing the spatial distribution of $CO_2$ and $N_2O$ emissions from soils. The RF classification may solely originate from the hypothesis and the structure of the STEP-GENDEC-$N_2O$ model if we were working at a local scale. However, since we are operating at a regional scale and the data inputted into the RF model reflect the spatial distribution of the explored factors in the region, we can attribute the RF classification, our result (Fig. 6), to a combination of the STEP-GENDEC-$N_2O$ model's structure and the specific biophysical/edaphic conditions prevalent in the Sahelian band under investigation.

In the literature, soil water content is often highlighted as the major driver of the temporal variation of soil $N_2O$ emissions as it regulates the oxygen availability to soil microbes (Butterbach-Bahl et al., 2013; Davidson and Verchot, 2000). The effect of soil moisture is actually predominant on denitrification processes, which lead to large amounts of $N_2O$ emissions when soil water-filled pore space (WFPS) reaches 70 to 80% (Davidson and Verchot, 2000). This is consistent with the result of our RF analysis, which ranks soil water content as the second most important factor responsible for spatial changes in soil $N_2O$ emissions (Fig. 6b). The impact of air temperature and soil temperature on the spatial distribution of soil $CO_2$ emissions suggests a positive feedback loop between climate warming and these emissions. The impact of global change drivers such as temperature on ecosystem processes and greenhouse gas emissions has been well studied and proven (Aulakh et al., 1992; Bajracharya et al., 2000; Lloyd and Taylor, 1994; Ray et al., 2020). The annual budgets of $CO_2$ and $N_2O$ emissions (Fig. 7a and 7b) throughout the period of simulation show a low interannual variability. This can be attributed to the low interannual variability of influencing factors such as substrate availability (C: $33.60 \pm 2.38$ gC m$^{-2}$ d$^{-1}$ N: $5.89 \pm 0.46$ gN m$^{-2}$ d$^{-1}$), and soil water content ($4.87 \pm 0.19\%$ yr$^{-1}$). Our simulation results do not allow us to explore possible interactions between climate warming and annual soil $CO_2$ and $N_2O$ emissions, as the average annual air temperature (averaged over the study domain) did not vary much over the simulation period ($28.37 \pm 0.25$ °C). Regional-scale observations show a temperature increase ranging from 1 to 2°C between 1950 and 2010 (Guichard et al., 2020). Therefore, over a 10-year period, this corresponds to a maximum increase of approximately 0.33°C, which is less than 0.5°C. This order of magnitude is comparable to the one computed for air temperature from the climate dataset used, and it is too small to be detected by temperature-versus-time regression.

## 4.4 Uncertainties and limitations

The lack of comprehensive dataset on the annual spatial distribution and growth dynamics of the livestock population in the Sahel between 2012 and 2022 remains a significant source of uncertainty in the $CO_2$ and $N_2O$ emissions reported in this study. Actually, information on the spatial distribution and population of livestock was only available for the 2010 year (Gilbert et al., 2018). Only the spatial and the seasonal variability of the grazing pressure was taken into account in our simulation. We assumed that the annual distribution and growth dynamics of livestock in Sahelian SPSs did not change significantly between

2010 and 2022, although it might have been affected by the interannual variability of herbaceous mass. Given the significant impact of livestock on $CO_2$ and $N_2O$ emissions in these ecosystems (Agbohessou et al., 2023; Assouma et al., 2017; Soussana et al., 2010; Valentini et al., 2014), an increase in livestock population during the study period could result in the misestimation of soil $CO_2$ and $N_2O$ emissions. Significant changes in the spatial distribution of animal load from one year to another could also lead to some uncertainties in the simulated spatial distribution of the emissions. Furthermore, it's worth noting that our

estimate does not account for tree root respiration, which can lead to an underestimate of the total soil $CO_2$ emissions in regions with high tree density.

In a previous study employing the STEP-GENDEC-$N_2O$ model at the local scale (within a silvopastoral system located in Senegal), Agbohessou et al. (2023) conducted an uncertainty analysis for STEP-GENDEC-$N_2O$ using a Monte Carlo simulation and a sensitivity analysis with Sobol's method (Sobol, 2001). In this study, they evaluated the overall uncertainty

surrounding $CO_2$ and $N_2O$ emissions simulated by STEP-GENDEC-$N_2O$ and identified the key parameters/variables to which the $CO_2$ and $N_2O$ emissions simulated by STEP-GENDEC-$N_2O$ are most sensitive. They found that the $CO_2$ and $N_2O$ emissions simulated by STEP-GENDEC-$N_2O$ at the local scale are particularly sensitive to soil texture. This being the case, another significant source of uncertainty in the $CO_2$ and $N_2O$ emissions reported in this study arises from the accuracy of the different input datasets used, especially the soil and precipitation datasets. We used the best dataset available for our region (to our

knowledge) for all input variables. However, the accuracy of our estimate also depends on the accuracy of the input dataset used. The choice of the various input datasets in this study is based on expert recommendations, comparison of the results of uncertainty analyses conducted for the different datasets in their respective reference articles, and their availability for our study region.

The soil C and N contents are significant factors influencing the spatial distribution of soil $CO_2$ and $N_2O$ emissions in Sahelian

SPSs, as indicated by our RF analysis. However, despite the availability of some local measurement data (Elberling et al., 2003a, b) and databases related to soil C and N content (Hengl et al., 2021) in the Sahel region, accurately assessing the temporal variability of these elements in Sahelian SPSs soils remains challenging.

Finally, we assumed that natural or anthropogenic disturbances such as wildfires in Sahelian SPSs during our simulation period are fairly negligible. Uncertainties related to disturbances like wildfire are actually difficult to estimate as there are varying

perspectives and conflicting findings in the literature regarding the impact of burning on $N_2O$ emissions (Karhu et al., 2015; Takakai et al., 2006).

## 5 Conclusions and perspectives

Our study advances the understanding of the spatial distribution and annual budget of $CO_2$ and $N_2O$ emissions from soil in the Sahel. Information on the magnitude of $CO_2$ and $N_2O$ emissions from soils in underrepresented areas are actually important to
shed light on the contribution of these areas to the overall GHGs budget and thereby inform the development of effective mitigation strategies that can help reduce GHG emissions. SPSs represent a significant portion of the West African drylands, where they have expanded due to global warming and are expected to continue expanding in the near future (Thornton and Herrero, 2015). Previous studies at the local scale in the Sahel have shown that soils in semi-arid ecosystems are notable contributors to GHGs emissions (Assouma et al., 2017; Brümmer et al., 2009; Delon et al., 2017). Our results extended these
local estimates to a broader spatiotemporal scale, showing that, overall, Sahelian SPSs soil emits less $CO_2$ and $N_2O$ than tropical areas and croplands, on a global scale. Furthermore, by mapping emissions we provided crucial insights into the localization of soil $CO_2$ and $N_2O$ emission hotspots, thereby offering indirect assessments of soil health in the Sahel region. This information can be a valuable asset for land managers who can leverage it to devise and implement effective strategies aimed at minimizing emissions and fostering carbon sequestration.

To further refine estimates of soil $CO_2$ and $N_2O$ emissions in Sahelian SPSs, efforts to collect comprehensive datasets on livestock spatial distribution and temporal dynamics, tree densities and fire are needed. Additionally, more experimental studies should investigate the roles of nitrification and denitrification processes for soil $N_2O$ emissions and the role of the decomposition process for $CO_2$ emissions in semi-arid ecosystems to better parameterize the model.

## 6 Appendices

**Appendix A:**

**Figure A1: Spatial distribution of silvopastoral areas in the Sahel. (Details on how the percentage of silvopastoral area pixels within the simulation grid cells were computed are provided in Methodology, Session 2.4)**

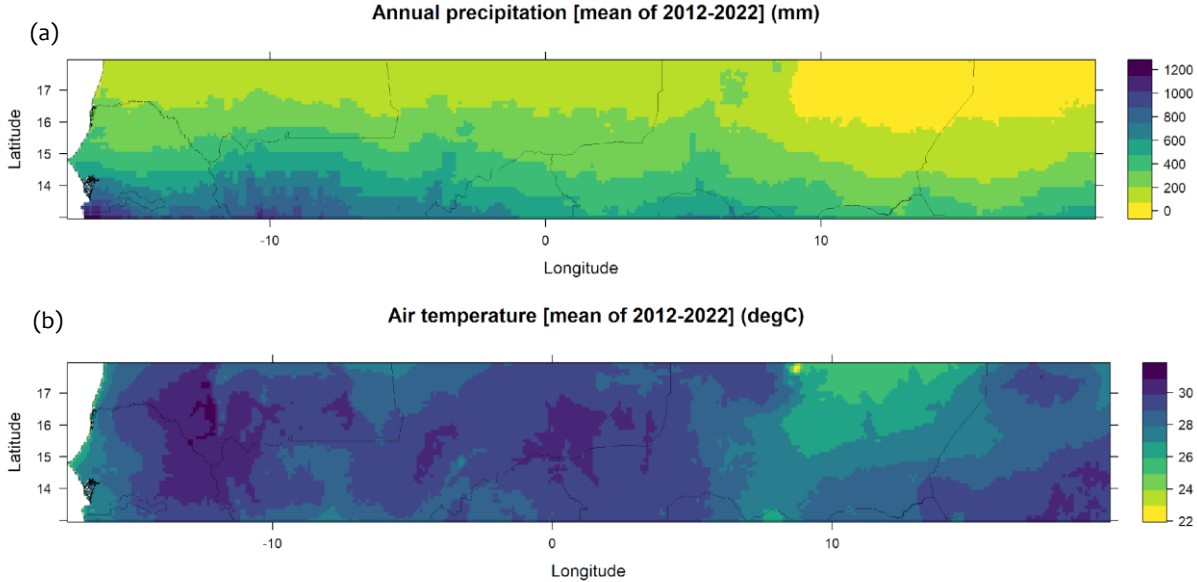

**Figure A2: Spatial distribution of precipitation and air temperature (mean over 2012-2022).**

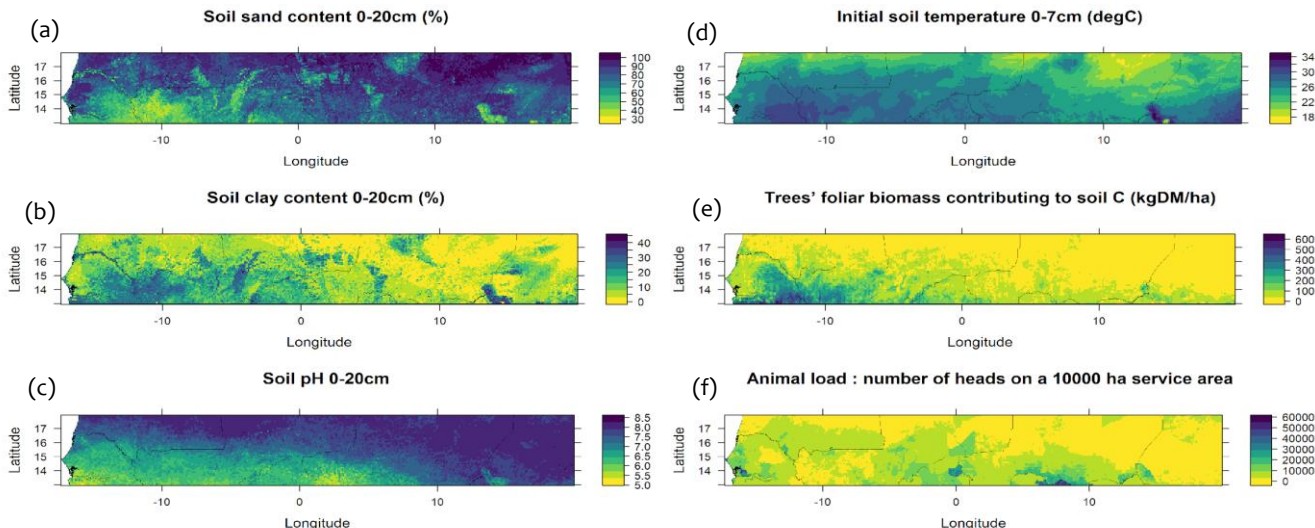

**Figure A3: Spatial distribution of soil properties, trees' foliar biomass and livestock**



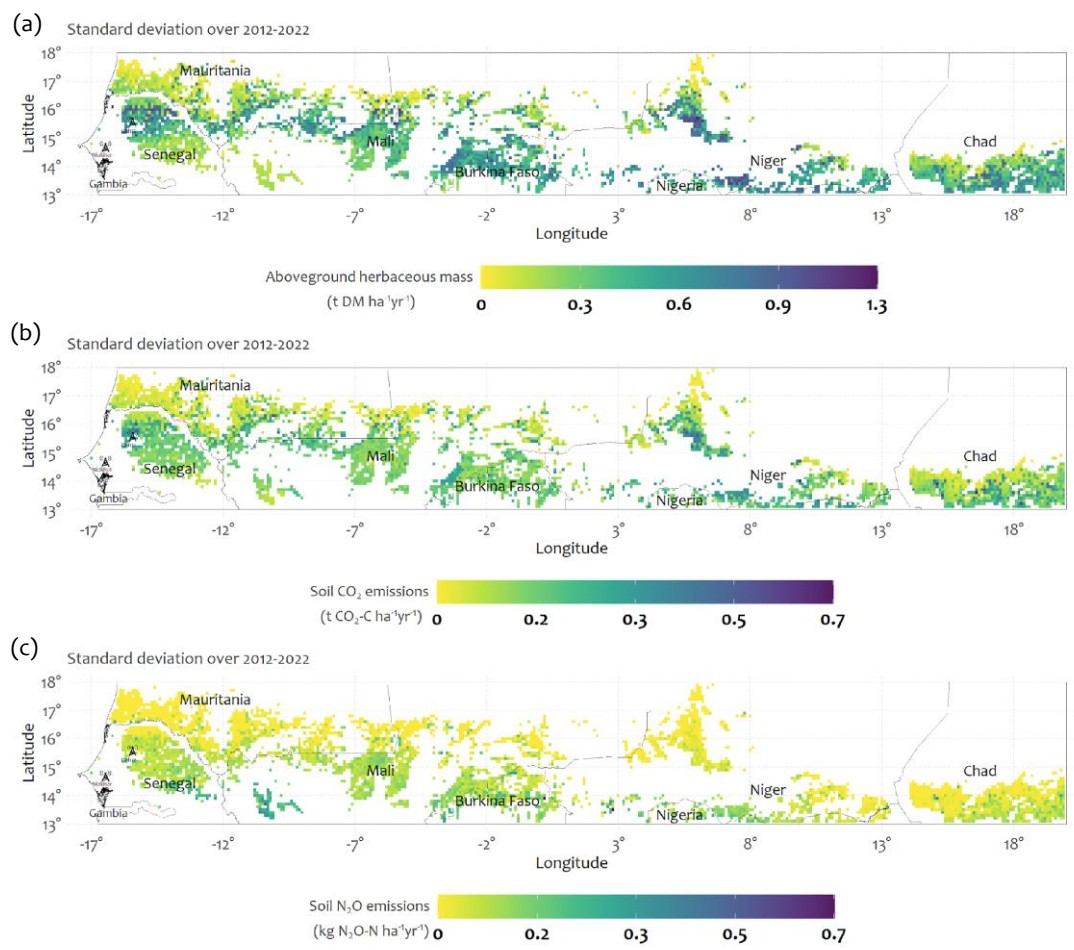

**Figure A4: Standard deviations of the spatial distribution of (a) herbaceous biomass, (b) soil CO$_2$ and (c) soil N$_2$O emissions in Sahelian SPSs (over 2012-2022). Only pixels dominated by SPSs (>80%) are displayed**

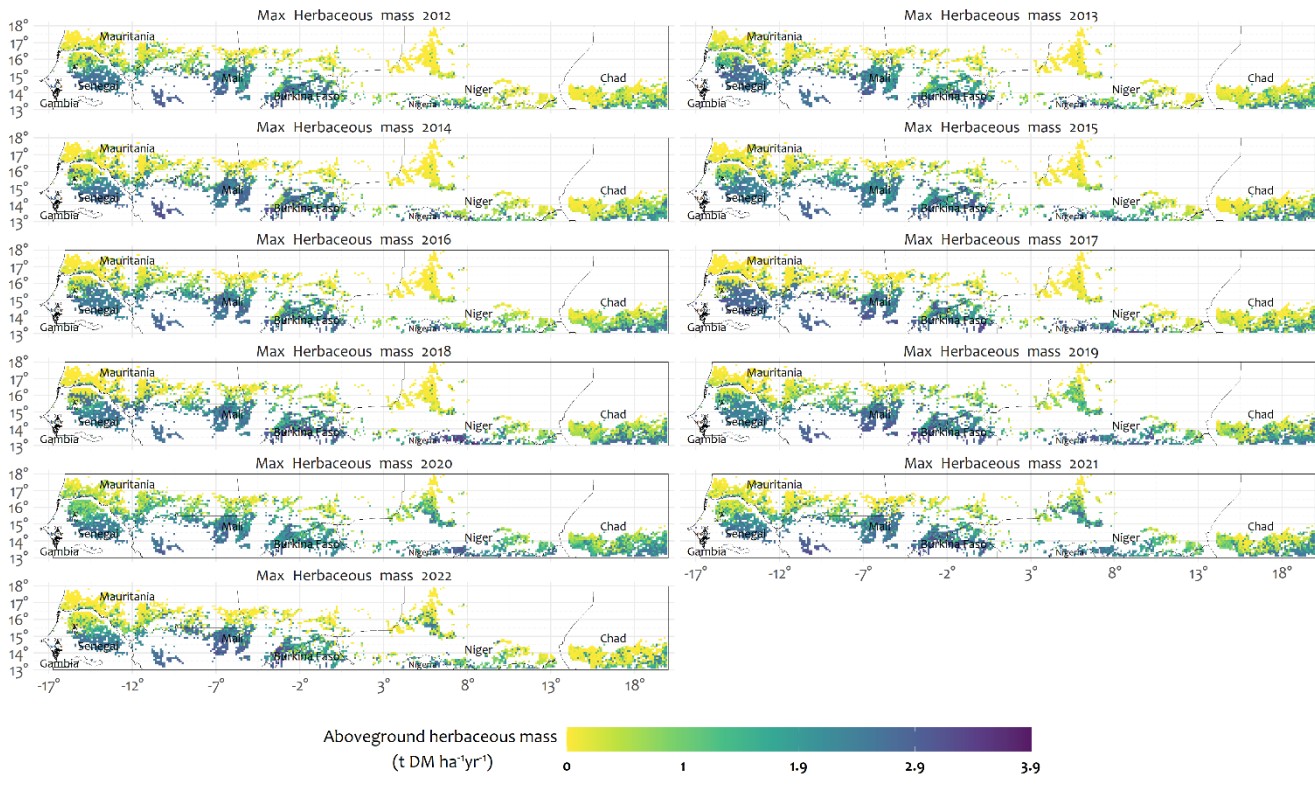


**Figure A5: Annual spatial distribution of herbaceous biomass in Sahelian SPSs (2012-2022). Only pixels dominated by SPSs (>80%) are displayed**

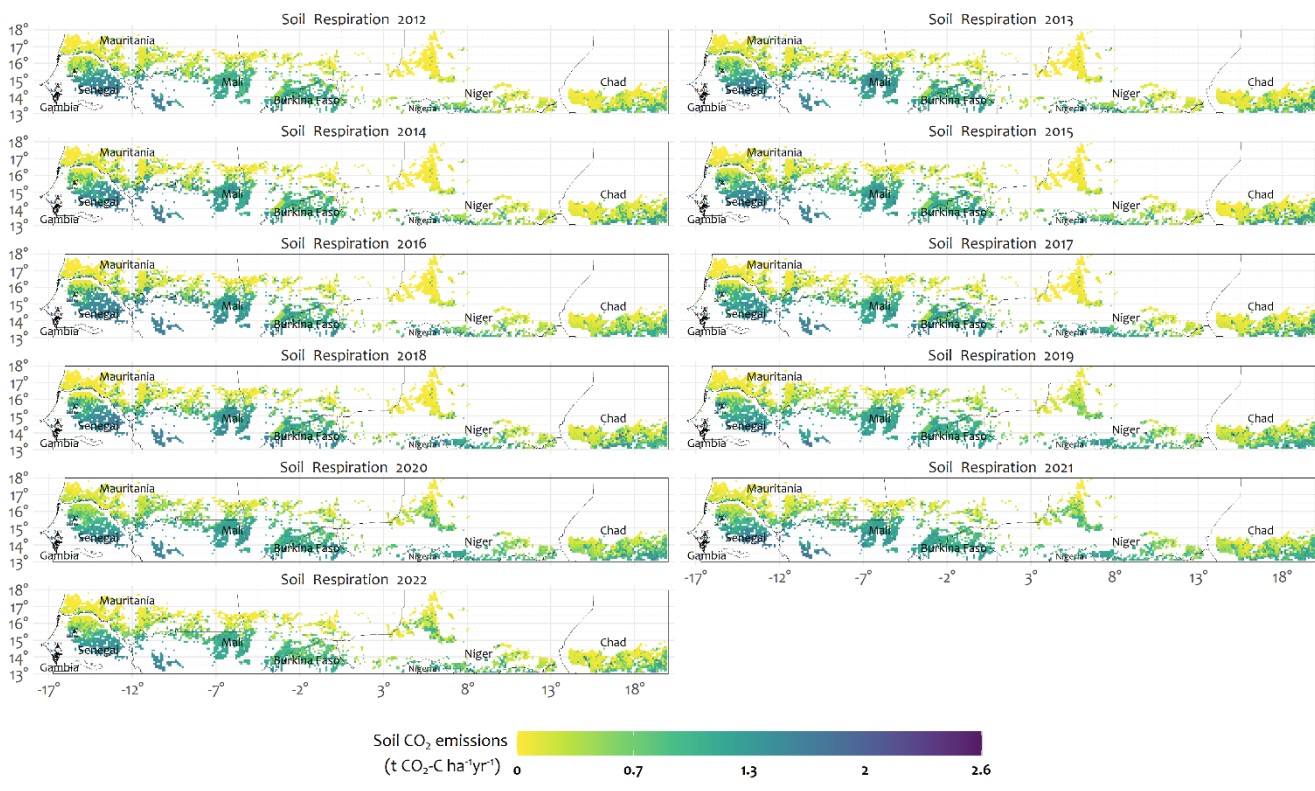

Figure A6: Annual spatial distribution of soil CO₂ emissions in Sahelian SPSs (2012-2022). Only pixels dominated by SPSs (>80%) are displayed

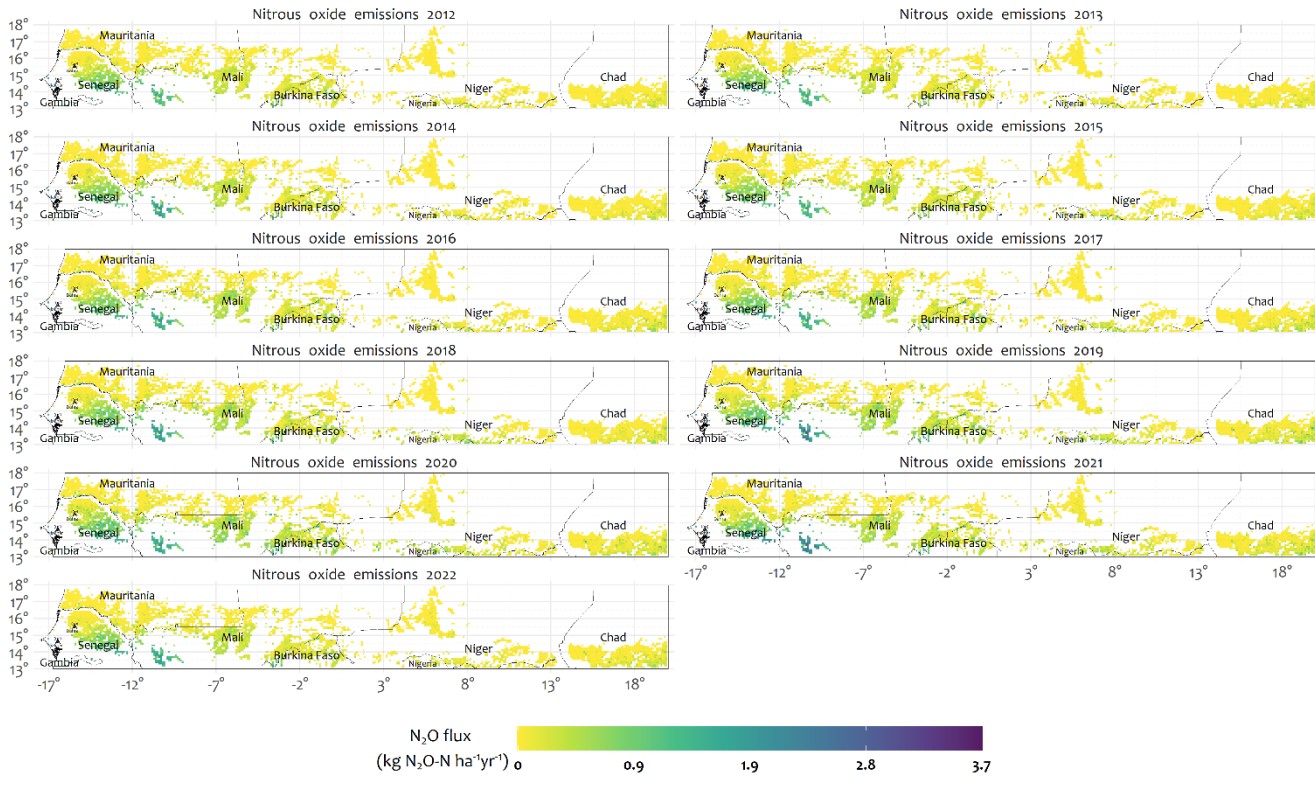

**Figure A7: Annual spatial distribution of soil N₂O emissions in Sahelian SPSs (2012-2022). Only pixels dominated by SPSs (>80%) are displayed**

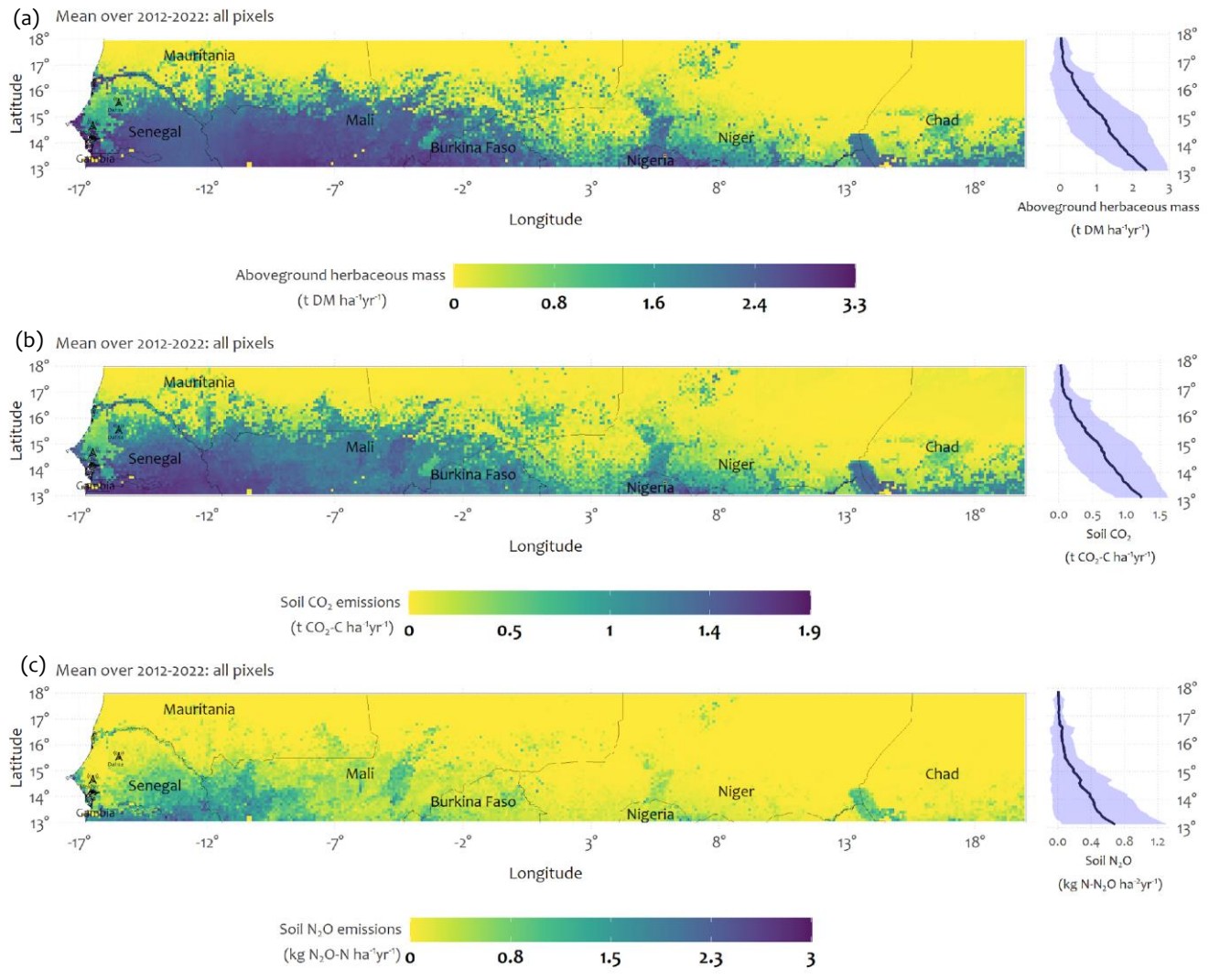

**Figure A8: Regional distribution of simulated (a) herbaceous biomass, (b) soil CO₂ and (c) soil N₂O emissions in Sahelian SPSs (annual mean over 2012-2022). All pixels are displayed. The right panel shows (a) herbaceous biomass, (b) soil CO₂ and (c) soil N₂O emissions along a latitudinal gradient of 0.1°, while the shaded area indicates the standard deviation.**

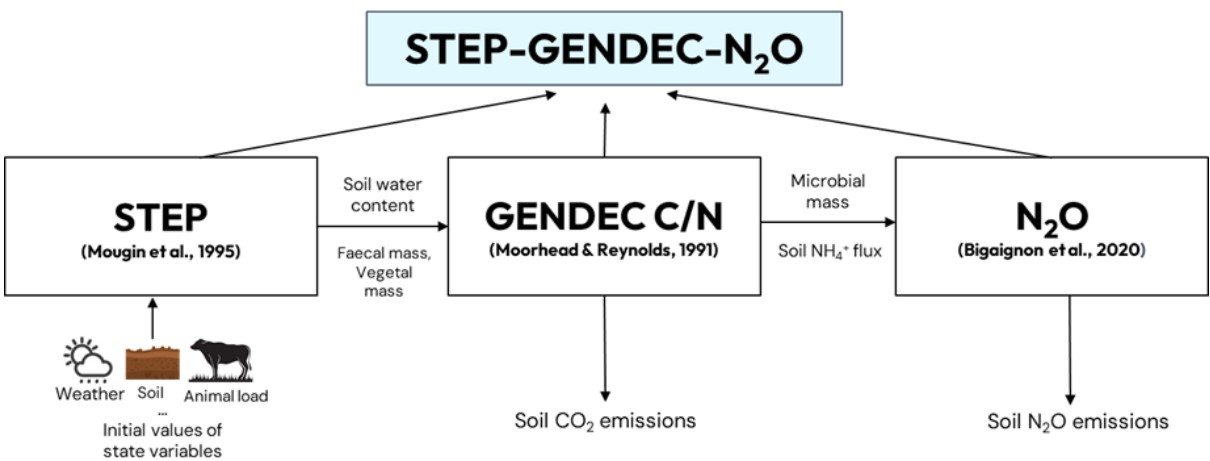


**Figure A9: Summary figure showing the connection between the models STEP, GENDEC, and the N₂O module.**

**Code availability.** The 2D STEP-GENDEC-N₂O model is available on Zenodo at https://doi.org/10.5281/zenodo.7866671. The "rstep" R package (Agbohessou, 2022), developed to automate workflows for 1D and 2D STEP-GENDEC-N₂O simulations, has been archived on Zenodo at: https://doi.org/10.5281/zenodo.7994028.

**Author contribution.** YA, CD, MG, EM, and OR conceived and designed the study. EM, CD, MG, and YA developed the 2D STEP-GENDEC-N₂O model code and YA performed the simulations. YA prepared the manuscript with contributions from all co-authors.

**Competing interests.** The authors declare that they have no conflict of interest.

**Funding.** This work was supported by the "Carbon sequestration and greenhouse gas emissions in (agro) silvopastoral 630     ecosystems in the Sahelian CILSS states" (CaSSECS) project (FOOD/2019/410-169), which received funding from the European Union under the "Development of Smart Innovation through Research in Agriculture" (DeSIRA) Initiative. Additionally, this research received financial support from the "Intergrated Nitrogen Studies in Africa" (INSA) project, which is supported by the European Union's Horizon 2020 Research and Innovation programme under the Marie Skłodowska-Curie Actions (RISE MSCA) Grant Agreement n° 871944.

**Mention.** This document has been produced with the assistance of the European Union. Its content is the sole responsibility of the authors and can in no way be taken to reflect the position of the European Union.

**Acknowledgments.** The authors gratefully acknowledge the use of the Nuwa high-performance computing cluster from Toulouse-France, which provided the essential computing resources that facilitated the research presented in this paper.

Additionally, the authors wish to express their appreciation to Anne Mottet for sharing relevant literature on land cover

products.

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
