# Peer review of "Modelling CO2 and N2O emissions from soils in silvopastoral systems of the West-African Sahelian band"

_EGUsphere, 2023_

## Author Response (AR3)

**Modelling $CO_2$ and $N_2O$ emissions from soils in silvopastoral systems of the West-African Sahelian band**

**Agbohessou et al.**

**Authors' Responses to Referee's comments**

*We express our gratitude to the referee for their invaluable comments, which we have taken into consideration and addressed. We hope that these revisions will lead to an improved manuscript. In the following responses, we will be referencing the line numbers of the version of the manuscript that the reviewers have provided comments and questions on. Our responses are in italic.*

**RC= Referee's Comment** *and AR = Authors' Response*

**2nd Revision**

**RC:** thank you very much for the revised version of the manuscript. It has been improved quite a lot. However the referee and myself agree on the fact that the length of the spinup still needs some clarification.

Please submit a revised version with better explanations on the spinup procedure.

*AR: Thank you, we provided more clarification on the spin-up time at lines 142-145. Now these lines read: "Simulations were performed over an 11-year period (2012-2022) preceded by a 6-year spin-up using the meteorological forcing data of year 2012, which was repeated 6 times. The spin up period allows for carbon and nitrogen pools to reach stability, as in Agbohessou et al. (2023). Indeed, in the model, the carbon compartments for buried litter, feces, and dry roots are not initialized at 0, thus our simulations start with initial carbon values of 3.7, 0.3, and 6.0 gC for buried litter, feces, and dry roots, respectively. These values represent means derived from in situ measurements collected over several years at the Dahra site, where the model has been previously employed at the local scale. The carbon and nitrogen submodel used is relatively simple, employing first-order differential equations with moderate nonlinearity, which likely accounts for the rapid convergence observed in the model. All this explains why extensive spin-up time is not necessary for running the model with appropriately supplied carbon and nitrogen compartments."*

**1st Revision**

**RC1: Anonymous Referee #1, 12 Feb 2024**

**General comment**

**RC:** The present research presents an application of the STEP-GENDEC-$N_2O$ model at the macro regional level of the Sylvo Pastoral Systems in the West-African Sahelian band at high spatial resolution. The results are revealing that soil $CO_2$ and $N_2O$ emission densities in these systems and regions are very low compared to other region and to cropland systems in general. An analysis of the factors controlling the spatial distribution of $CO_2$ and $N_2O$ emissions is conducted using a random forest model. This approach, however, is quite sophisticated and prevent the author from developing a more hypothesis-driven approach, which would have been more appropriated here. Indeed, this complex statistical method result in rather obvious results, which might actually be rather due to the model equation than to the specificity of the systems investigated here (see my comments below). This kind of approach also deprives

scientists from developing intelligible intuitions that can be tested through experimental procedure or logical reasoning.

*AR: We used the RF model to explore the factors which influence $CO_2$ and $N_2O$ emissions. Our goal was to determine the extent to which the factors impact the spatial variability of $CO_2$ and $N_2O$ emissions. The Random Forest (RF) model, in fact, offers a means to rank each variable by their importance in explaining the spatial distribution of $CO_2$ and $N_2O$ emissions.*
*We provide a more detailed response to this comment below.*

RC: In addition, the uncertainty analysis of the present research misses an important point: the uncertainty associated to the input datasets used (see my comment below).

*AR: We agree that the uncertainties associated with input data need to be more detailed in the manuscript. We provide a more detailed response to this comment below.*

**Abstract**

RC: Authors investigate the $CO_2$ and $N_2O$ emissions in sylvo-pastoral systems in the Sahel using a process based-model. The first question that comes to mind is what about the $CO_2$ and $N_2O$ budget. As you mentioned that "spatial variations in soil $CO_2$ emissions are primarily controlled by soil carbon content, temperature, herbaceous mass and animal load", I am inclined to assume that high $CO_2$ emissions might also be associated with high C sink (by SOC and herbaceous mass). If authors indeed investigated $CO_2$ emissions and not "net $CO_2$ emissions", then it is useful to point out this limitation from the abstract.

*AR: Thank you for your observation. Indeed, our study focused solely on $CO_2$ emissions, not "net $CO_2$ emissions." As suggested, we have clarified this distinction in the abstract from the beginning by modifying the sentence at line 19:"In this work, we spatialized and applied the process-based model STEP-GENDEC-$N_2O$ to investigate the magnitude, spatial, and temporal patterns of herbaceous mass production, as well as $CO_2$ and $N_2O$ emissions (not net emissions) from soil in Sahelian SPS"*

**Introduction**

RC: Lines 37-38: Even though emissions from terrestrial soils played a major role in the C emissions since the 1700s, it played the "primary role" only in the period 1700-1930/50 but then fossil combustion became the main drivers. It would be fairer to explicit this temporal trend in GHG emissions since 1700.

*AR: We followed the suggestion by modifying the sentence at lines 37-38. Now these lines read: "The atmospheric concentrations of $CO_2$ and $N_2O$ have experienced significant increases since the late 1700s (Bloch-Johnson et al., 2021; Prinn et al., 2018). This surge is primarily attributed to emissions originating from terrestrial soils (Butterbach-Bahl et al., 2013; Chevallier et al., 2015; Tian et al., 2020) during the period from 1700 to 1980 (Kammen and Mario, 1993). However, post-1990, the major contributors to greenhouse gas emissions on a global scale shifted to the energy systems and industrial sectors (Parmesan et al., 2022)."*

RC: Line 50: Do you have a reference to support this statement?

*AR: Yes, here are the references:*
- *Le Houerou 1987 : Le Houerou, Henry N. "Indigenous shrubs and trees in the silvopastoral systems of Africa." Agroforestry (1987): 139. https://citeseerx.ist.psu.edu/document?repid=rep1&type=pdf&doi=66a9638f37bdcdb 68a484271e807a4434748fdcd#page=148*
- *Turner et al., 2014 also shows that pastoralism remains a prevalent way of life in semi-arid zones (such as the sahelian SPS) : Turner M D, McPeak J G, Ayantunde A. 2014.*

*The role of livestock mobility in the livelihood strategies of rural peoples in semi-arid West Africa. Human Ecology, 42(2): 231–247.*
- *Herrero et al., 2013, doi:10.1017/S1751731112001954 (Livestock production in the developing world occurs in a wide range of heterogeneous production systems. These can range from pastoral/grassland-based systems, which occupy most of the land area and have low human population densities)*
- *Herrero et al., 2013, doi:10.1073/pnas.1308149110 (Biomass use, production, feed efficiencies, and greenhouse gas emissions from global livestock systems)*

*We added these references to line 50*

**RC:** Lines 75-80: Another limitation to process-based model is the lack of reliable and accurate input datasets to the model. For instance, pedological properties are often associated to high uncertainties, even in well-documented regions such as Europe (see Ballabio et al. https://doi.org/10.1016/j.geoderma.2015.07.006).

**AR:** *We agree that input datasets especially regarding soil properties are a major issue for process-based approaches at the large scale. This point is further discussed below. We have added the following sentence to line 80: "Additionally, reliable and accurate large spatial scale input datasets for the process-based models are often lacking, not only in under-represented areas but also in well-documented regions like Europe (Ballabio et al., 2016)"*

**RC:** Lines 85-89: It is a good point to know that the model used has been developed for Sahelian semi-arid ecosystems and validated against several sites of Sahelian SPS but it would also be useful to know how well did the model perform in these previous studies (or maybe this is already discussed in the discussion section?).

**AR:** *We included in the discussion at line 372 how well the model performed in the previous studies. Line 372 now reads: "In this study, we have up-scaled the 1D STEP-GENDEC-$N_2O$ model, which was previously used in local studies across various sites in the West Sahel region. For example, in previous studies conducted at a SSP located in the northern region of Senegal (Dahra: 15°24'10''N, 15°25'56''W), Bigaignon et al. (2020) effectively used STEP-GENDEC-$N_2O$ to satisfactorily simulate soil water content ($R^2 = 0.68$ & RMSE = 1.67 mm d$^{-1}$), $NO_3^-$ content in soil ($R^2 = 0.42$ & RMSE = 0.83 mgN kgsoil$^{-1}$), and $N_2O$ emissions ($R^2 = 0.36$ & RMSE = 2.51 ng N m$^{-2}$ s$^{-1}$). At the same site, Agbohessou et al. (2023) successfully simulated $CO_2$ fluxes using STEP-GENDEC-$N_2O$ combined with a tree growth model (DynACof), achieving convincing results for Gross Primary Productivity (GPP: EF = 0.49 & RMSE = 2.15 gC m$^{-2}$ d$^{-1}$) and Ecosystem Respiration (Reco: EF = 0.56 & RMSE = 1.34 gC m$^{-2}$ d$^{-1}$). Additionally, Delon et al. (2019) demonstrated successful simulation of soil respiration on the identical site using STEP-GENDEC-$N_2O$. On another SSP located in Mali (Agoufou: 15.34° N, 1.48° W), Delon et al. (2015) employed STEP-GENDEC-$N_2O$ to satisfactorily simulate soil moisture ($R^2 = 0.7$), soil temperature ($R^2 = 0.86$), and herbaceous phytomass ($R^2 = 0.72$), yielding satisfactory results. Building upon these previous local applications and validations of the STEP-GENDEC-$N_2O$ model in different representative sites of the Sahelian SPS, to provide the first large-scale estimate of soil $CO_2$ and $N_2O$ emissions from W-Sahelian SPS."*

**RC:** Lines 81-92: Overall, at this stage, it remains a bit confusing whether the modelling approach provides an estimate of $CO_2$ and N2O net emissions (i.e., budget) or only estimates $N_2O$/$CO_2$ emissions.

**AR:** *As recommended, we have clarified from the beginning in the abstract that we are solely estimating $CO_2$ and $N_2O$ emissions from soil, not net emissions.*

**Methods**

**RC:** Lines 136: I am a bit surprised that a 6-year spin-up is enough to reach C and N pools stability. Are you sure about that duration?

**AR:** *Yes. In the model, the carbon compartments for buried litter, feces and dry roots are not set at 0, so our simulations do not start with 0 carbon/nitrogen. This explains why it does not require a lot of years of spin-up to run the model with a properly supplied carbon/nitrogen compartments. We have added this clarification to the manuscript at line 137.*

**RC:** Lines 169-173: Although the authors are likely using the best available datasets, they are still associated with important uncertainties. For instance, the soil water content in the ERA5 datasets is calculated using the HTessel model, which uses a soil water budget to estimate runoff, evaporation and soil water content. The soil water content depends on the evaporation calculated based on a modified Penman accounting for the aerodynamic, stomatal and soil hydric stress resistances and on the soil water capacity which is derived from a very simple soil texture map and root depth map provided by the FAO. Therefore, the soil texture map used to derive the soil water content is not the same as the one used by the authors in the present study. This might result in inconsistencies for the soil biogeochemistry module.

**AR:** *We agree that using soil water content data (the temporal dynamic) and soil texture data from two separate datasets for model simulation may introduce inconsistencies within the soil biogeochemistry module. However, it is worth mentioning that for our simulation, we only needed to initialize the soil water content at the beginning (first time step only). The model inherently simulates the temporal dynamics of soil water content for the entire simulation period based on the input soil texture (e.g., In STEP-GENDEC-N$_2$O, the soil water is distributed within soil layers according to a tipping bucket scheme and soil water contents at field capacity and at the wilting point are derived as a function of soil texture.). This resolves the inconsistency issue that could arise if we were to use as model input temporal dynamics of soil water content from a different dataset (along with soil texture data from another dataset) instead of simulating it based on the soil texture, as we have done.*

*Moreover, the capability of global LSMs, including HTESSE, and STEP, to reproduce soil fluxes (ETR, soil moisture and runoff) in the study region have been analysed in previous model intercomparison exercises (e.g., Boone et al. (2009), Grippa et al. (2011), Grippa et al. (2017)). Among the 20 different LSMs studied in Grippa et al. (2017) HTESSEL was among those performing the best. The methodology employed by HTESSEL with a variable infiltration rate that considers the integrated soil moisture in the top 50cm was indeed found to work well in the Sahel. In Grippa et al. (2017), they have also assessed the sensitivity of LSMs to input soil maps, comparing model performances when using a specific map derived for Gourma area (Northern Mali) and the ECOCLIMAP2 soil map. they have found that employing as input a soil map that explicitly represents the soils in the study area improved the representation of water fluxes by some LSMs, i.e. those having the physical processes or parameterizations to correctly take into account surface hydrology over these kinds of areas.*

*References:*
- *M. Grippa, L. Kergoat, A. Boone, C. Peugeot, J. Demarty, B. Cappelaere, L. Gal, P. Hiernaux, E. Mougin, A. Ducharne, E. Dutra, C. Hain, M. Anderson and the ALMIP2 Working group (2017) "Modelling surface runoff and water fluxes over contrasted soils in pastoral Sahel: evaluation of the ALMIP2 land surface models over the Gourma region in Mali" J. Hydrometeorology, 18, 7, doi: 10.1175/JHM-D-16-0170.1*
- *M. Grippa, L. Kergoat,F. Frappart,Q. Araud,A. Boone,P. De Rosnay,J.-M. Lemoine, S. Gascoin, G. Balsamo, C. Ottlé, B. Decharme, S.Saux-Picart and G. Ramillien (2011) « Land water storage changes over West Africa estimated by GRACE and land surface models » Wat. Res. Res 47 , W05549.*
- *A. Boone, P. de Rosnay, G. Balsamo, A. Beljaars, F. Chopin, B. Decharme, C. Delire, A. Ducharne, S. Gascoin, M. Grippa, F. Guichard, Y. Gusev, P. Harris, L. Jarlan, L. Kergoat, E. Mougin, O. Nasonova, A. Norgaard, T. Orgeval, C. Ottlé, I. Poccard-Leclercq, J. Polcher, I. Sandholt, S. Saux-Picart, C. Taylor (2009). « The AMMA Land Surface Model Intercomparison Project (ALMIP) ».Bull. Amer. Meteor. Soc., 90(12), 1865-1880*

*We clarified this by modifying the sentence at line 172. Now lines 172-173 read: "We initialized the dry soil albedo, soil moisture (mm), and soil temperature (°C) at the beginning of the simulation (first time step only) using data extracted from the ECMWF re-Analysis ERA5_Land (Muñoz Sabater, 2019)."*

**RC:** More generally the soil datasets, at this level of resolution, comes with high inaccuracies (again see Fig. 1 in Ballabio et al. 2016). I'm not saying this invalidate the results of the present research but the uncertainties associated to these high resolutions soil datasets require a presentation (how high are the uncertainties? Were they even established in previous studies?) and a discussion at some point in the manuscript (could be in the method presentation or in the discussion section).

***AR:*** *Thank you for your comment. We agree that the uncertainties associated with input data need to be more detailed in the manuscript. To address this, we have included additional details in the paragraph describing the soil dataset used (iSDA soil). Specifically, we have provided information regarding the performance and uncertainties of the model used to predict iSDA soil data. We added to line 173, the following sentences: "Prediction uncertainty estimates per pixel for the iSDA soil properties are provided in Hengl et al. (2021). In the same study, the average accuracy performance based on fivefold spatial cross-validation for various soil variables indicated that soil pH exhibited the highest performance with a concordance correlation coefficient (CCC) of 0.90. The CCC values for soil clay content, sand content, and silt content were 0.85, 0.85, and 0.78, respectively."*
*The model performance and individual variable importance lists can be tracked via the https://zenodo.org. See the links below for soil properties*
- *iSDAsoil: soil pH for Africa predicted at 30 m resolution at 0-20 and 20-50 cm depths (https://zenodo.org/records/4220290)*
- *iSDAsoil: soil sand content (USDA system) for Africa predicted at 30 m resolution at 0-20 and 20-50 cm depths (https://zenodo.org/records/4094607)*
- *iSDAsoil: soil clay content (USDA system) for Africa predicted at 30 m resolution at 0-20 and 20-50 cm depths (https://zenodo.org/records/4085160)*

**RC:** Line 184-185: I would presume that the FAO provide an estimate of the livestock population (at the national level) in the 2012-2020 period?

***AR:*** *Yes, FAO provides estimates of the livestock population at the national level for the period from 2012 to 2020. However, these data are only available at the national scale and have not been downscaled to the finer spatial scales required for our simulation.*
*GLW 3 is currently the most recently compiled and harmonized subnational livestock distribution data available (and only covers the year 2010). In Gilbert et al. (2018) it is mentioned that the outputs of the GLW 3 dataset have been adjusted to ensure that the total number of animals in a country aligns with the FAOSTAT 2010 total stock (Gilbert et al., 2018). There are no recent datasets available prior to 2010 presenting livestock distribution at the subnational scale in our region to our knowledge.*
*We have added this clarification to the manuscript at line 185.*

*Reference:*
- *Gilbert, M., Nicolas, G., Cinardi, G., Van Boeckel, T. P., Vanwambeke, S. O., Wint, G. R. W., and Robinson, T. P.: Global 665 distribution data for cattle, buffaloes, horses, sheep, goats, pigs, chickens and ducks in 2010, Sci Data, 5, 180227, https://doi.org/10.1038/sdata.2018.227, 2018.*

**RC:** Lines 180-193 and 195-220: Same remark than for the soil datasets: What are the uncertainties associated to these input datasets? The reason why I am picky with the uncertainties of the input datasets is because the study is a model application so the interpretation of the results will strongly depend on the accuracy and reliability of the input datasets used.

*AR:* *To discuss the uncertainties related to the soil dataset, we added to line 173, the following sentences: "Prediction uncertainty estimates per pixel for the iSDA soil properties are provided in Hengl et al. (2021). In the same study, the average accuracy performance based on fivefold spatial cross-validation for various soil variables indicated that soil pH exhibited the highest performance with a concordance correlation coefficient (CCC) of 0.90. The CCC values for soil clay content, sand content, and silt content were 0.85, 0.85, and 0.78, respectively."*

**Results**

**RC:** Sections 3.1 and 3.2: So, it appears from these results that high $CO_2$ and high $N_2O$ emissions are observed in more or less where we can find the highest herbaceous production. Again, the question that comes into my mind is did you estimate the C sink in the soil and vegetation? It is quite logical that area with very little vegetation will also have very little SOC stock and soil micro-organisms but then these regions will also have very small C sink in soil and vegetation. By contrast, tropical and equatorial ecosystems have very high $CO_2$ emissions (due to soils and vegetation respiration) but also very high $CO_2$ adsorption by photosynthesis. Therefore, the most important question is not the emissions but the net emissions by soils and vegetations.

*AR:* *We acknowledge the interest in estimating the spatial distribution of net $CO_2$ emissions across the region, achieving this requires simulating $CO_2$ fluxes (photosynthesis and autotrophic respiration) from trees across the region. Recently, a product developed by Tucker et al. (2023) provides an estimate of tree population across the Sahel as well as carbon content in roots, leaves, and trunks. However, this information alone is insufficient. To simulate $CO_2$ flux from trees across the Sahel, we require more comprehensive data on the trees, such as species information and phenology, at a regional scale. Unfortunately, such a dataset is not currently available for the Sahel region to our knowledge. Additionally, we would need to couple our model with a tree growth model like DynACof (Vezy et al., 2020), which is not yet spatialized. This actually presents an interesting opportunity for future research that deserves exploration. Given these limitations associated with simulating the tree component at a regional scale, for this study, we have chosen to concentrate exclusively on investigating $CO_2$ emissions from soil, at least for the time being.*

*References:*
  - *Tucker, C., Brandt, M., Hiernaux, P., Kariryaa, A., Rasmussen, K., Small, J., Igel, C., Reiner, F., Melocik, K., Meyer, J., Sinno, S., Romero, E., Glennie, E., Fitts, Y., Morin, A., Pinzon, J., McClain, D., Morin, P., Porter, C., Loeffler, S., Kergoat, L., Issoufou, B.-A., Savadogo, P., Wigneron, J.-P., Poulter, B., Ciais, P., Kaufmann, R., Myneni, R., Saatchi, S., and Fensholt, R.: Sub-continental-scale carbon stocks of individual trees in African drylands, Nature, 615, 80–86, https://doi.org/10.1038/s41586-022-05653-6, 2023.*
  - *Vezy, R., le Maire, G., Christina, M., Georgiou, S., Imbach, P., Hidalgo, H.G., Alfaro, E.J., Blitz-Frayret, C., Charbonnier, F., Lehner, P., Loustau, D., Roupsard, O., 2020. DynACof: a process-based model to study growth, yield and ecosystem services of coffee agroforestry systems. Environ. Model. Softw. 124, 104609 https://doi.org/10.1016/j.envsoft.2019.104609.*

**RC:** Lines 336-338: How do you explain the result that the spatial distributions of $CO_2$ and $N_2O$ emissions are primarily controlled by the spatial distributions of soil carbon and nitrogen contents? To be clearer I am wondering to which extent this result should be attributed to the hypotheses and structure of the model used in the present research or to the specific biophysical and edaphic conditions of the Sahelian band investigated here? Maybe this question is answered later in the discussion.

*AR: We used the RF model to also explore factors beyond SOC and SON, which influence $CO_2$ and $N_2O$ emissions. Our goal was to determine also the extent to which other variables impact the spatial variability of $CO_2$ and $N_2O$ emissions.*

*It is true that in the STEP-GENDEC-$N_2O$ model a certain number of variables are used to simulate $CO_2$ and $N_2O$ emissions. However, simply examining the model's structure does not allow us to discern the extent to which each variable impacts the model output at a broader spatial scale. The Random Forest (RF) model, in fact, offers a means to rank each variable by their importance in explaining the spatial distribution of $CO_2$ and $N_2O$ emissions. This can also be interpreted as a way to identify the sensitivity of the model (at larger spatial scale) to different input variables involved in calculating $CO_2$ and $N_2O$ emissions.*

*Given that we are operating at a regional scale and the data inputted into the RF model reflect the spatial distribution of the explored variables (or factors) in the region, we can attribute the RF classification (our result) to a combination of the model's structure and the specific biophysical/edaphic conditions of the Sahelian band under investigation.*

**Discussion**

**RC:** Line 451-453: Author find that C and N soil content is the main factor controlling the spatial distribution of C and N emissions from soils. This is very logical if the SOC model used is a first-order kinetic model (i.e., SOC and SON mineralizations are proportional to the stocks) modified by some pedoclimatic parameters such as soil humidity, temperature, soil texture (which are the factors that come right after SOC and SON content in explaining the spatial distribution of C and N emissions). So, I am wondering if a sophisticated random forest modelling approaches is needed to demonstrate some thing that looks like a circular reasoning to me (the model assumptions involve first order kinetic decomposition of SOC and SON, findings are that SOC and SON drive the spatial distribution of C and N emissions). Please, try to better differentiate what is a result expected from the model equations and which are the results that arise from the particular conditions of sylvo pastoral systems in the Sahel to avoid circular reasoning.

*AR: Indeed, the results brought by the Random Forest (RF) approach confirm what we could have expected so the RF approach brings an objective and necessary confirmation. These results indicate that the model equations are adapted and suitable to the particular conditions of silvopastoral systems explored. The RF gives results from a combined approach of influencing parameters and gives a ranking of these parameters.*

*Given that we are operating at a regional scale and the data given as input to the RF model reflect the spatial distribution of the explored variables (or factors) in the region, we can attribute the RF classification (our result) to a combination of the model's structure and the specific biophysical/edaphic conditions of the Sahelian band under investigation.*

*We clarified this in the discussion at line 459.*

**RC:** Lines 478-497: A major uncertainty that need to be discussed is the one arising from the soil input datasets (see my comment above). I understand this is not an easy task but uncertainties, but do you have any idea of how uncertainties could be quantified (e.g., using Monte Carlo analysis using the uncertainty range of the input data used?).

*AR: Thank you for this interesting comment. Yes, we understand how uncertainties can be quantified, and we have already done so for the STEP-GENDEC-$N_2O$ model in our previous study at the local scale (within a silvopastoral system located in Senegal). In that study, we conducted an uncertainty analysis using a Monte Carlo simulation and a sensitivity analysis employing Sobol's method (Sobol, 2001) (see supplementary material of Agbohessou et al., 2023). In that study, we evaluated the overall uncertainty surrounding $CO_2$ and $N_2O$ emissions simulated by STEP-GENDEC-$N_2O$ and identified the key parameters/variables to which the $CO_2$ and $N_2O$ emissions simulated by STEP-GENDEC-$N_2O$ are most sensitive. You can read more about how these uncertainty and sensitivity analysis were performed for the STEP-*

*GENDEC-N₂O model in the supplementary material of our previous paper published in open access in Agricultural and Forest Meteorology. (Agbohessou et al. 2023:* *https://doi.org/10.1016/j.agrformet.2023.109780**).*

*The results of this sensitivity analysis at the local scale show that the $CO_2$ and $N_2O$ emissions simulated by STEP-GENDEC-$N_2O$ are most sensitive to soil texture (see the figure below).*

[Figure]

*This agrees with your point, suggesting that soil input data are important and should be discussed more extensively for their uncertainties. Therefore, we have added the following text to the discussion at line 490: "In a previous study employing the STEP-GENDEC-$N_2O$ model at the local scale (within a silvopastoral system located in Senegal), Agbohessou et al. (2023) conducted an uncertainty analysis for STEP-GENDEC-$N_2O$ using a Monte Carlo simulation and a sensitivity analysis with Sobol's method (Sobol, 2001). In this study, they evaluated the overall uncertainty surrounding $CO_2$ and $N_2O$ emissions simulated by STEP-GENDEC-$N_2O$ and identified the key parameters/variables to which the $CO_2$ and $N_2O$ emissions simulated by STEP-GENDEC-$N_2O$ are most sensitive. They found that the $CO_2$ and $N_2O$ emissions simulated by STEP-GENDEC-$N_2O$ at the local scale are particularly sensitive to soil texture. This being the case, another significant source of uncertainty in the $CO_2$ and $N_2O$ emissions reported in this study arises from the accuracy of the different input datasets used, especially the soil and precipitation datasets. We used the best available dataset for our region (to our knowledge) for all input variables. However, the accuracy of our estimate also depends on the accuracy of the input dataset used. The choice of the various input datasets in this study is based on expert recommendations, comparison of the results of uncertainty analyses conducted for the different datasets in their respective reference articles, and their availability for our study region."*

*References:*
- *Sobol, I.M., 2001. Global sensitivity indices for nonlinear mathematical models and their Monte Carlo estimates. Mathematics and computers in simulation 55, 271–280.*
- *Agbohessou et al. 2023. To what extent are greenhouse-gas emissions offset by trees in a Sahelian silvopastoral system? https://doi.org/10.1016/j.agrformet.2023.109780*

RC: Even though the model used has been validated in previous studies, another major uncertainty still concerns the lack of comparison of the model results against site observations.

Indeed, the previous model validations were performed at the site level, using site level input data whose quality might be much better than the regional and global datasets used here. Since model outputs uncertainties are not only related to model equations and parameters (structural uncertainties) but also to the quality of the input datasets (operational uncertainties) and since the present research is mostly an application of an existing model to a large regional scale, a possible way to evaluate the uncertainties associated to the input datasets could be to compare the simulation performances of the model using locally measured input datasets for the two validation sites (as provided in previous studies) versus the simulation performances of the model using regional and global input datasets in the pixels of the two validation sites.

*AR: Yes, in previous studies, the outputs of the STEP-GENDEC-N$_2$O model at the local scale have been compared to local measurements (see Agbohessou et al. (2023), Delon et al. (2019), Delon et al. (2015), and Bigaignon et al. (2020)). However, we did not compare our regional simulation outputs (pixel values) to local measurements. We also did not compare them to previous simulation outputs at the local scale because our regional simulation has a resolution of about 123.21 km². This type of comparison (regional simulation pixel values to local measurements) raises significant scaling issues since local measurements are not representative of the larger scale. For example, the picture below showcases the spatial heterogeneity of NDVI (Sentinel-2) within one of our simulation pixels covering a site located in Senegal where the model has been locally applied. We have validated the orders of magnitude of the simulated fluxes compared to previous studies.*

[Figure]

References:

- Bigaignon, L., Delon, C., Ndiaye, O., Galy-Lacaux, C., Serça, D., Guérin, F., Tallec, T., Merbold, L., Tagesson, T., Fensholt, R., André, S., and Galliau, S.: Understanding N2O Emissions in African Ecosystems: Assessments from a Semi-Arid Savanna Grassland in Senegal and Sub-Tropical Agricultural Fields in Kenya, 26, 2020.
- Delon, C., Mougin, E., Serça, D., Grippa, M., Hiernaux, P., Diawara, M., Galy-Lacaux, C., and Kergoat, L.: Modelling the effect of soil moisture and organic matter degradation on biogenic NO emissions from soils in Sahel rangeland (Mali), Biogeosciences, 12, 3253–3272, https://doi.org/10.5194/bg-12-3253-2015, 2015

- Delon, C., Galy-Lacaux, C., Serça, D., Personne, E., Mougin, E., Adon, M., Dantec, V. L., Loubet, B., Fensholt, R., and Tagesson, T.: Modelling land–atmosphere daily exchanges of NO, NH3, and CO2 in a semi-arid grazed ecosystem in Senegal, 29, 2019
- Agbohessou et al. 2023. To what extent are greenhouse-gas emissions offset by trees in a Sahelian silvopastoral system? https://doi.org/10.1016/j.agrformet.2023.109780

**RC2: Bertrand Guenet, 28 Feb 2024**

**RC:** Before I begin my review, I should point out that I am the editor of this paper and that I had a lot of trouble finding a second reviewer, so to avoid making you wait too long I decided to do the review myself.

This paper presents the results of modelling work aimed at simulating CO2 and N20 fluxes in the western Sahel using a new model combining STEP, GENDEC and DNDC. The study focuses particularly on areas with silvopastoral systems.

The study is well written and interesting. In addition, it responds to an important need in a relatively unknown geographical area, and it develops an interesting methodology that is potentially applicable to other geographical areas. Nevertheless, the lack of evaluation of the model's outputs is a major flaw in the study, and I feel that it is essential to provide elements for evaluating the model. There are a number of soil respiration products (Warner et al. 2019, for example) or above-ground biomass that should be compared at least with the model outputs.

*AR: Thank you for your suggestion. We have read the paper by Warner et al. (2019) and downloaded their product for comparison with our simulated soil $CO_2$ emissions (see the graph below) as suggested.*

[Figure]

*Some key points that need to be highlighted:*
- *Warner et al. (2019) products have a resolution of 0.008333 x 0.008333 degrees, whereas our simulation has a resolution of 0.1 x 0.1 degrees. Before comparison, we calculated the mean soil respiration predicted by Warner et al. (2019) in our simulation grid cells. It is important to highlight that the Warner et al. (2019) product is a one-time estimate*

*based on input data from 01-01-1963, to 31-12- 2011, while our simulated soil $CO_2$ emissions used for comparison represent the annual mean of the period from 2012 to 2022.*

- *We used a process-based model, while the soil $CO_2$ emissions (soil respiration) predicted by Warner et al. (2019) are based on a machine learning approach, quantile regression forest model. This model was trained using selected environmental predictors and 2,657 input soil respiration observations from the SRDB (The global Rs database, SRDB; Bond-Lamberty & Thomson, 2010b). However, it's important to note that the SRDB database used by Warner et al. (2019) does not contain measurements from sites located in our region (the simulation area). There are only two measurement sites in West Africa for soil respiration (see figure below from Warner et al. (2019)). Additionally, Warner et al. (2019) mentioned that the greatest prediction uncertainties were observed in semi-arid ecosystems.*

[Figure]

*We have discussed this comparison in the discussion at line 418.*

*As suggested we have also compared the aboveground herbaceous mass (ABG) simulated by STEP-GENDEC-N₂O with the ABG biomass product from ACF (see figure below)*

[Figure]

*Fig: The relationship between aboveground (ABG) herbaceous mass simulated by STEP-GENDEC-N2O and ABG biomass predicted by ACF (Bernard and Fillol, 2021, 2020; Lambert*

*et al., 2019). Each point represents the annual mean biomass in a simulation pixel. The dashed line represents the 1:1 line, while the solid line depicts the linear regression line.*

*We added this figure and the following text to the discussion at line 390: "We compared the aboveground herbaceous mass (ABG) simulated by STEP-GENDEC-N₂O with the ABG biomass product from ACF for SPS pixels only. This revealed a significant correlation between the ABG herbaceous mass simulated by STEP-GENDEC-N₂O and the ACF biomass product (with an R² value of 0.61 and RMSE of 1.51). The ABG biomass derived from ACF amounts to 7 tDM ha⁻¹ yr⁻¹, whereas the simulated ABG herbaceous mass by STEP-GENDEC-N₂O does not exceed 3 tDM ha⁻¹ yr⁻¹. This variation can be attributed to the ACF product being derived from satellite data, encompassing not only herbaceous plants but also including the tree and crop component within these SPS-dominated pixels. Additionally, the Monteith formulation used by ACF approaches a potential biomass and therefore corresponds more to the upper bound of the STEP-GENDEC-N₂O simulations."*

*STEP alone was already compared to remote sensing index for green and dry vegetation (NDVI and STI respectively) in a previous work by Pierre et al. 2016 (see figures below)*

[Figure]

[Figure]

*Figures 1 and 2 in Pierre et al (2016).*

*Fig. 1. Mean values over 2001–2014 of: (a) total simulated vegetation mass (g.m−2) at green vegetation maximum; (b) total simulated vegetation mass (g·m−2) averaged over October–June; (c) MODIS iNDVI; and (d) MODIS STI averaged over October–June; (e) is the topsoil sand proportion (%) of the dominant soil unit according to HWSD.*

*Fig.2 Regional-mean cycles of vegetation components simulated by STEP (top) and MODIS vegetation indices (bottom) for 2001–2014. Continuous (resp. dotted) lines correspond to PV (resp. PV\*) simulations. For each date, the reported value is the mean of the 14 regional means over the study area.*

**RC:** In addition, I have a couple of comments that need to be answered before publication. How interactive is the coupling between the models (STEP GENDEC, DNDC)? This is not detailed enough, is there feedbacks between modules? Are all the modules run together or one after the other to get the boundaries conditions?

*AR: Here is a summary figure showing the connection between the models STEP, GENDEC, and the $N_2O$ module. The models run one after another at every time step. The GENDEC (General Decomposition) model calculates the N and C content in the soil and the subsequent soil CO2 emissions, and the $N_2O$ module (based on DNDC processes) calculates the $N_2O$ emissions from GENDEC outputs. There are no feedbacks from the $N_2O$ module to GENDEC and STEP. The details can be found in Delon et al. (2015) for the links between STEP and GENDEC, and in Bigaigon et al. (2020) for the links between GENDEC and the $N_2O$ module. We have included the figure in the appendices and referenced it in the manuscript at line 129.*

[Figure]

**RC:** L123: What do you mean exactly by "inputs"? Boundary condition of the model?

*AR: Here we are referring to the input variables. The model uses these variables to calculate microbial respiration. This clarification is provided in the manuscript at line 123.*

**RC:** L136: I don't understand how 6 years could be enough to reach equilibrium for soil C and N, is the resolution of the spinup analytical? please provide more information.

*AR: In the model, the carbon compartments for buried litter, feces and dry roots are not set at 0, so our simulations do not start with 0 carbon/nitrogen. This explains why it does not require a lot of years of spin-up to run the model with a properly supplied carbon/nitrogen compartments. We have added this clarification to the manuscript at line 137.*

**RC:** Section 2.3.1: I don't understand why you used different products for climate data? By doing so there is a risk of inconstancies between the data use for precipitation and those used for the other climate variables. If really necessary this need to be better justified, if not I suggest to use the precipitation data from AgERA5.

*AR: The decision to use the IMERG dataset for precipitation over the ERA5 dataset is based on expert recommendations and a comparison of results from previous evaluations of both datasets in our region (Lavers et al. (2022), Coz et Giesen (2020)). This decision is explained in the manuscript by incorporating the following statement into section 2.3.1, at line 151: "Additionally, the choice of the Imerg dataset over the ERA5 dataset for precipitation is based on expert recommendations and the results of previous evaluations of ERA5 precipitation data by Lavers et al. (2022). Their study highlighted significant errors primarily in tropical regions. According to Lavers et al. (2022), users can only have confidence in ERA5 precipitation data in extratropical regions."*

*Reference:*
- *Lavers et al. (2022), An evaluation of ERA5 precipitation for climate monitoring https://doi.org/10.1002/qj.4351*
- *Camille Le Coz and Nick van de Giesen 2020. Comparison of Rainfall Products over Sub-Saharan Africa. https://doi.org/10.1175/JHM-D-18-0256.1*

**RC:** Section 2.5: I don't understand here what are the explaining variables you used with the RF, is it the STEP-GENDEC-N2O model outputs? Needs to be better explain.

*AR: The explanatory variables include the spatial distribution of various environmental and biological factors susceptible to impact the spatial distribution of the soil $CO_2$ and $N_2O$ emissions simulated by the STEP-GENDEC-$N_2O$ model. These factors consist of a combination of output variables from the STEP-GENDEC-$N_2O$ model (e.g., soil water content, soil*

*temperature, soil C content, soil N content and herbaceous mass) and input variables for the STEP-GENDEC-N₂O model (e.g., soil sand content, soil clay content, soil pH, air temperature, albedo, annual precipitation and animal load). We clarified this by modifying the sentences at lines 251-254*

**RC:** Section 3.2.4: What would be the difference with a Tier1 approach? Since you mention this limitation of using Tier 1 in the introduction a comparison with a Tier1 approach would be very informative.

**AR:** *We acknowledge that comparing our results to FAO tier 1 estimates could be very informative. However, we did not do such a comparison because FAO tier 1 data are provided at the national scale, whereas our simulations were conducted at the scale of the Sahel region, encompassing parts of various Sahelian countries. Additionally, downsizing FAO tier 1 data to match the specific territories within our simulation area would present considerable challenges. Nonetheless, in our discussion, we did compare our findings with those of other studies conducted at regional and global scales.*